# NASI: Label- and Data-agnostic Neural Architecture Search at Initialization

**Yao Shu, Shaofeng Cai, Zhongxiang Dai, Beng Chin Ooi & Bryan Kian Hsiang Low**
Department of Computer Science, National University of Singapore
{shuyao,shaofeng,daizhongxiang,ooibc,lowkh}@comp.nus.edu.sg

## Abstract

Recent years have witnessed a surging interest in *Neural Architecture Search* (NAS). Various algorithms have been proposed to improve the search efficiency and the search effectiveness of NAS, i.e., to reduce the search cost and improve the generalization performance of the selected architectures, respectively. However, the search efficiency of these algorithms is severely limited by the need for model training during the search process. To overcome this limitation, we propose a novel NAS algorithm called *NAS at Initialization* (NASI) that exploits the capability of a *Neural Tangent Kernel* in being able to characterize the performance of candidate architectures at initialization, hence allowing model training to be completely avoided to boost the search efficiency. Besides the improved search efficiency, NASI also achieves competitive search effectiveness on various datasets like CIFAR-10/100 and ImageNet. Further, NASI is shown to be *label-* and *data-agnostic* under mild conditions, which guarantees the transferability of the architectures selected by our NASI over different datasets.

## 1 Introduction

The past decade has witnessed the wide success of *deep neural networks* (DNNs) in computer vision and natural language processing. These DNNs, e.g., VGG (Simonyan & Zisserman, 2015), ResNet (He et al., 2016), and MobileNet (Howard et al., 2017), are typically handcrafted by human experts with considerable trials and errors. The human efforts devoting to the design of these DNNs are, however, not affordable nor scalable due to an increasing demand of customizing DNNs for different tasks. To reduce such human efforts, *Neural Architecture Search* (NAS) (Zoph & Le, 2017) has recently been introduced to automate the design of DNNs. As summarized in (Elsken et al., 2019), NAS conventionally consists of a search space, a search algorithm, and a performance evaluation. Specifically, the search algorithm aims to select the best-performing neural architecture from the search space based on its evaluated performance via the performance evaluation. In the literature, various search algorithms (Luo et al., 2018; Zoph et al., 2018; Real et al., 2019) have been proposed to search for architectures with comparable or even better performance than the handcrafted ones.

However, these NAS algorithms are inefficient due to the requirement of model training for numerous candidate architectures during the search process. To improve the search inefficiency, one-shot NAS algorithms (Pham et al., 2018; Dong & Yang, 2019; Liu et al., 2019; Xie et al., 2019) have trained a single one-shot architecture and then evaluated the performance of candidate architectures with model parameters inherited from this fine-tuned one-shot architecture. So, these algorithms can considerably reduce the cost of model training in NAS, but still require the training of the one-shot architecture. This naturally begs the question *whether NAS is realizable at initialization such that model training can be completely avoided during the search process?* To the best of our knowledge, only a few empirical efforts to date have been devoted to developing NAS algorithms without model training (Mellor et al., 2021; Park et al., 2020; Abdelfattah et al., 2021; Chen et al., 2021).

This paper presents a novel NAS algorithm called *NAS at Initialization* (NASI) that can completely avoid model training to boost search efficiency. To achieve this, NASI exploits the capability of a *Neural Tangent Kernel* (NTK) (Jacot et al., 2018; Lee et al., 2019a) in being able to formally characterize the performance of infinite-wide DNNs at initialization, hence allowing the performance of candidate architectures to be estimated and realizing NAS at initialization. Specifically, given

the estimated performance of candidate architectures by NTK, NAS can be reformulated into an optimization problem without model training (Sec. 3.1). However, NTK is prohibitively costly to evaluate. Fortunately, we can approximate it[1] using a similar form to gradient flow (Wang et al., 2020) (Sec. 3.2). This results in a reformulated NAS problem that can be solved efficiently by a gradient-based algorithm via additional relaxation using Gumbel-Softmax (Jang et al., 2017) (Sec. 3.3). Interestingly, NASI is even shown to be *label-* and *data-agnostic* under mild conditions, which thus implies the transferability of the architectures selected by NASI over different datasets (Sec. 4).

We will firstly empirically demonstrate the improved search efficiency and the competitive search effectiveness achieved by NASI in NAS-Bench-1Shot1 (Zela et al., 2020b) (Sec. 5.1). Compared with other NAS algorithms, NASI incurs the smallest search cost while preserving the competitive performance of its selected architectures. Meanwhile, the architectures selected by NASI from the DARTS (Liu et al., 2019) search space over CIFAR-10 consistently enjoy the competitive or even outperformed performance when evaluated on different benchmark datasets, e.g., CIFAR-10/100 and ImageNet (Sec. 5.2), indicating the guaranteed transferability of architectures selected by our NASI. In Sec. 5.3, NASI is further demonstrated to be able to select well-performing architectures on CIFAR-10 even with randomly generated labels or data, which strongly supports the *label-* and *data-agnostic* search and therefore the guaranteed transferability achieved by our NASI.

## 2 RELATED WORKS AND BACKGROUND

### 2.1 NEURAL ARCHITECTURE SEARCH

A growing body of NAS algorithms have been proposed in the literature (Zoph & Le, 2017; Liu et al., 2018; Luo et al., 2018; Zoph et al., 2018; Real et al., 2019) to automate the design of neural architectures. However, scaling existing NAS algorithms to large datasets is notoriously hard. Recently, attention has thus been shifted to improving the search efficiency of NAS without sacrificing the generalization performance of its selected architectures. In particular, a one-shot architecture is introduced by Pham et al. (2018) to share model parameters among candidate architectures, thereby reducing the cost of model training substantially. Recent works (Chen et al., 2019; Dong & Yang, 2019; Liu et al., 2019; Xie et al., 2019; Chen & Hsieh, 2020; Chu et al., 2020) along this line have further formulated NAS as a continuous and differentiable optimization problem to yield efficient gradient-based solutions. These one-shot NAS algorithms have achieved considerable improvement in search efficiency. However, the model training of the one-shot architecture is still needed.

A number of algorithms are recently proposed to estimate the performance of candidate architectures without model training. For example, Mellor et al. (2021) have explored the correlation between the divergence of linear maps induced by data points at initialization and the performance of candidate architectures heuristically. Meanwhile, Park et al. (2020) have approximated the performance of candidate architectures by the performance of their corresponding Neural Network Gaussian Process (NNGP) with only initialized model parameters, which is yet computationally costly. Abdelfattah et al. (2021) have investigated several training-free proxies to rank candidate architectures in the search space, while Chen et al. (2021) intuitively adopt theoretical aspects in deep networks (e.g., NTK (Jacot et al., 2018) and linear regions of deep networks (Raghu et al., 2017)) to select architectures with a good trade-off between its trainability and expressivity. Our NASI significantly advances this line of work in (a) providing theoretically grounded performance estimation by NTK (compared with (Mellor et al., 2021; Abdelfattah et al., 2021; Chen et al., 2021)), (b) guaranteeing the transferability of its selected architectures with its provable label- and data-agnostic search under mild conditions (compared with (Mellor et al., 2021; Park et al., 2020; Abdelfattah et al., 2021; Chen et al., 2021))) and (c) achieving SOTA performance in a large search space over various benchmark datasets (compared with (Mellor et al., 2021; Park et al., 2020; Abdelfattah et al., 2021)). Based on our results, Shu et al. (2022) recently have provided further improvement on existing training-free NAS algorithms.

### 2.2 NEURAL TANGENT KERNEL (NTK)

Let a dataset $(\mathcal{X}, \mathcal{Y})$ denote a pair comprising a set $\mathcal{X}$ of $m$ $n_0$-dimensional vectors of input features and a vector $\mathcal{Y} \in \mathbb{R}^{mn \times 1}$ concatenating the $m$ $n$-dimensional vectors of output values, respectively.

---

[1]More precisely, we approximate the trace norm of NTK.

Let a DNN be parameterized by $\boldsymbol{\theta}_t \in \mathbb{R}^p$ at time $t$ and output a vector $\boldsymbol{f}(\mathcal{X}; \boldsymbol{\theta}_t) \in \mathbb{R}^{mn \times 1}$ (abbreviated to $\boldsymbol{f}_t$) of the predicted values of $\mathcal{Y}$. Jacot et al. (2018) have revealed that the training dynamics of DNNs with gradient descent can be characterized by an NTK. Formally, define the NTK $\boldsymbol{\Theta}_t(\mathcal{X}, \mathcal{X}) \in \mathbb{R}^{mn \times mn}$ (abbreviated to $\boldsymbol{\Theta}_t$) as

$$\boldsymbol{\Theta}_t(\mathcal{X}, \mathcal{X}) \triangleq \nabla_{\boldsymbol{\theta}_t} \boldsymbol{f}(\mathcal{X}; \boldsymbol{\theta}_t) \nabla_{\boldsymbol{\theta}_t} \boldsymbol{f}(\mathcal{X}; \boldsymbol{\theta}_t)^\top . \tag{1}$$

Given a loss function $\mathcal{L}_t$ at time $t$ and a learning rate $\eta$, the training dynamics of the DNN can then be characterized as

$$\nabla_t \boldsymbol{f}_t = -\eta \, \boldsymbol{\Theta}_t(\mathcal{X}, \mathcal{X}) \, \nabla_{\boldsymbol{f}_t} \mathcal{L}_t, \quad \nabla_t \mathcal{L}_t = -\eta \, \nabla_{\boldsymbol{f}_t} \mathcal{L}_t^\top \, \boldsymbol{\Theta}_t(\mathcal{X}, \mathcal{X}) \, \nabla_{\boldsymbol{f}_t} \mathcal{L}_t . \tag{2}$$

Interestingly, as proven in (Jacot et al., 2018), the NTK stays asymptotically constant during the course of training as the width of DNNs goes to infinity. NTK at initialization (i.e., $\boldsymbol{\Theta}_0$) can thus characterize the training dynamics and also the performance of infinite-width DNNs.

Lee et al. (2019a) have further revealed that, for DNNs with over-parameterization, the aforementioned training dynamics can be governed by their first-order Taylor expansion (or linearization) at initialization. In particular, define

$$\boldsymbol{f}^{\mathrm{lin}}(\boldsymbol{x}; \boldsymbol{\theta}_t) \triangleq \boldsymbol{f}(\boldsymbol{x}; \boldsymbol{\theta}_0) + \nabla_{\boldsymbol{\theta}_0} \boldsymbol{f}(\boldsymbol{x}; \boldsymbol{\theta}_0)^\top (\boldsymbol{\theta}_t - \boldsymbol{\theta}_0) \tag{3}$$

for all $\boldsymbol{x} \in \mathcal{X}$. Then, $\boldsymbol{f}(\boldsymbol{x}; \boldsymbol{\theta}_t)$ and $\boldsymbol{f}^{\mathrm{lin}}(\boldsymbol{x}; \boldsymbol{\theta}_t)$ share similar training dynamics over time, as described formally in Appendix A.2. Besides, following the definition of NTK in (1), this linearization $\boldsymbol{f}^{\mathrm{lin}}$ achieves a constant NTK over time.

Given the mean squared error (MSE) loss defined as $\mathcal{L}_t \triangleq m^{-1} \|\mathcal{Y} - \boldsymbol{f}(\mathcal{X}; \boldsymbol{\theta}_t)\|_2^2$ and the constant NTK $\boldsymbol{\Theta}_t = \boldsymbol{\Theta}_0$, the loss dynamics in (2) above can be analyzed in a closed form while applying gradient descent with learning rate $\eta$ (Arora et al., 2019):

$$\mathcal{L}_t = m^{-1} \sum_{i=1}^{mn} (1 - \eta \lambda_i)^{2t} (\boldsymbol{u}_i^\top \mathcal{Y})^2 , \tag{4}$$

where $\boldsymbol{\Theta}_0 = \sum_{i=1}^{mn} \lambda_i(\boldsymbol{\Theta}_0) \boldsymbol{u}_i \boldsymbol{u}_i^\top$, and $\lambda_i(\boldsymbol{\Theta}_0)$ and $\boldsymbol{u}_i$ denote the $i$-th largest eigenvalue and the corresponding eigenvector of $\boldsymbol{\Theta}_0$, respectively.

## 3 NEURAL ARCHITECTURE SEARCH AT INITIALIZATION

### 3.1 REFORMULATING NAS VIA NTK

Given a loss function $\mathcal{L}$ and the model parameters $\boldsymbol{\theta}(\mathcal{A})$ of architecture $\mathcal{A}$, we denote the training and validation loss as $\mathcal{L}_{\mathrm{train}}$ and $\mathcal{L}_{\mathrm{val}}$, respectively. NAS is conventionally formulated as a bi-level optimization problem (Liu et al., 2019):

$$\min_{\mathcal{A}} \mathcal{L}_{\mathrm{val}}(\boldsymbol{\theta}^*(\mathcal{A}); \mathcal{A})$$
$$\text{s.t. } \boldsymbol{\theta}^*(\mathcal{A}) \triangleq \arg\min_{\boldsymbol{\theta}(\mathcal{A})} \mathcal{L}_{\mathrm{train}}(\boldsymbol{\theta}(\mathcal{A}); \mathcal{A}) . \tag{5}$$

Notably, model training is required to evaluate the validation performance of each candidate architecture in (5). The search efficiency of NAS algorithms (Real et al., 2019; Zoph et al., 2018) based on (5) is thus severely limited by the cost of model training for each candidate architecture. Though recent works (Pham et al., 2018) have considerably reduced this training cost by introducing a one-shot architecture for model parameter sharing, such a one-shot architecture still requires model training and hence incurs training cost.

To completely avoid this training cost, we exploit the capability of a NTK in characterizing the performance of DNNs at initialization. Specifically, Sec. 2.2 has revealed that the training dynamics of an over-parameterized DNN can be governed by its linearization at initialization. With the MSE loss, the training dynamics of such linearization are further determined by its constant NTK. Therefore, the training dynamics and hence the performance of a DNN can be characterized by the constant NTK of its linearization. However, this constant NTK is computationally costly to evaluate. To this end, we instead characterize the training dynamics (i.e., MSE) of DNNs in Proposition 1 using the trace norm of NTK at initialization, which fortunately can be efficiently approximated. For simplicity, we use this MSE loss in our analysis. Other widely adopted loss functions (e.g., cross entropy with softmax) can also be applied, as supported in our experiments. Note that throughout this paper, the parameterization and initialization of DNNs follow that of Jacot et al. (2018). For a $L$-layer DNN, we denote the output dimension of its hidden layers and the last layer as $n_1 = \cdots = n_{L-1} = k$ and $n_L = n$, respectively.

**Proposition 1.** *Suppose that $\|\boldsymbol{x}\|_2 \leq 1$ for all $\boldsymbol{x} \in \mathcal{X}$ and $\mathcal{Y} \in [0,1]^{mn}$ for a given dataset $(\mathcal{X}, \mathcal{Y})$ of size $|\mathcal{X}| = m$, a given $L$-layer neural architecture $\mathcal{A}$ outputs $\boldsymbol{f}_t \in [0,1]^{mn}$ as predicted labels of $\mathcal{Y}$ with the corresponding MSE loss $\mathcal{L}_t$, $\lambda_{\min}(\boldsymbol{\Theta}_0) > 0$ for the given NTK $\boldsymbol{\Theta}_0$ w.r.t. $\boldsymbol{f}_t$ at initialization, and gradient descent (or gradient flow) is applied with learning rate $\eta < \lambda_{\max}^{-1}(\boldsymbol{\Theta}_0)$. Then, for any $t \geq 0$, there exists a constant $c_0 > 0$ such that as $k \to \infty$,*

$$\mathcal{L}_t \leq mn^2(1 - \eta\overline{\lambda}(\boldsymbol{\Theta}_0))^q + \epsilon \tag{6}$$

*with probability arbitrarily close to 1 where $q$ is set to $2t$ if $t < 0.5$, and 1 otherwise, $\overline{\lambda}(\boldsymbol{\Theta}_0) \triangleq (mn)^{-1} \sum_{i=1}^{mn} \lambda_i(\boldsymbol{\Theta}_0)$, and $\epsilon \triangleq 2c_0\sqrt{n/(mk)}\left(1 + c_0\sqrt{1/k}\right)$.*

Its proof is in Appendix A.3. Proposition 1 implies that NAS can be realizable at initialization. Specifically, given a fixed and sufficiently large training budget $t$, in order to select the best-performing architecture, we can simply minimize the upper bound of $\mathcal{L}_t$ in (6) over all the candidate architectures in the search space. Since both theoretical (Mohri et al., 2018) and empirical (Hardt et al., 2016) justifications in the literature have shown that training and validation loss are generally highly related, we simply use $\mathcal{L}_t$ to approximate $\mathcal{L}_{\text{val}}$. Hence, (5) can be reformulated as

$$\min_{\mathcal{A}} \ mn^2(1 - \eta\overline{\lambda}(\boldsymbol{\Theta}_0(\mathcal{A}))) + \epsilon \quad \text{s.t.} \ \overline{\lambda}(\boldsymbol{\Theta}_0(\mathcal{A})) < \eta^{-1}. \tag{7}$$

Note that the constraint in (7) is derived from the condition $\eta < \lambda_{\max}^{-1}(\boldsymbol{\Theta}_0(\mathcal{A}))$ in Proposition 1, and $\eta$ and $\epsilon$ are typically constants[2] during the search process. Following the definition of trace norm, (7) can be further reduced into

$$\max_{\mathcal{A}} \|\boldsymbol{\Theta}_0(\mathcal{A})\|_{\text{tr}} \quad \text{s.t.} \ \|\boldsymbol{\Theta}_0(\mathcal{A})\|_{\text{tr}} < mn\eta^{-1}. \tag{8}$$

Notably, $\boldsymbol{\Theta}_0(\mathcal{A})$ only relies on the initialization of $\mathcal{A}$. So, no model training is required in optimizing (8), which achieves our objective of realizing NAS at initialization.

Furthermore, (8) suggests an interesting interpretation of NAS: NAS intends to select architectures with a good trade-off between their model complexity and the optimization behavior in their model training. Particularly, architectures containing more model parameters will usually achieve a larger $\|\boldsymbol{\Theta}_0(\mathcal{A})\|_{\text{tr}}$ according to the definition in (1), which hence provides an alternative to measuring the complexity of architectures. So, maximizing $\|\boldsymbol{\Theta}_0(\mathcal{A})\|_{\text{tr}}$ leads to architectures with large complexity and therefore strong representation power. On the other hand, the complexity of the selected architectures is limited by the constraint in (8) to ensure a well-behaved optimization with a large learning rate $\eta$ in their model training. By combining these two effects, the optimization of (8) naturally trades off between the complexity of the selected architectures and the optimization behavior in their model training for the best performance. Appendix C.1 will validate such trade-off. Interestingly, Chen et al. (2021) have revealed a similar insight of NAS to us.

## 3.2 Approximating the Trace Norm of NTK

The optimization of our reformulated NAS in (8) requires the evaluation of $\|\boldsymbol{\Theta}_0(\mathcal{A})\|_{\text{tr}}$ for each architecture $\mathcal{A}$ in the search space, which can be obtained by

$$\|\boldsymbol{\Theta}_0(\mathcal{A})\|_{\text{tr}} = \sum_{\boldsymbol{x} \in \mathcal{X}} \left\|\nabla_{\boldsymbol{\theta}_0(\mathcal{A})} \boldsymbol{f}(\boldsymbol{x}, \boldsymbol{\theta}_0(\mathcal{A}))\right\|_{\text{F}}^2, \tag{9}$$

where $\|\cdot\|_{\text{F}}$ denotes the Frobenius norm. However, the Frobenius norm of the Jacobian matrix in (9) is costly to evaluate. So, we propose to approximate this term. Specifically, given a $\gamma$-Lipschitz continuous loss function[3] $\mathcal{L}_{\boldsymbol{x}}$ (i.e., $\|\nabla_{\boldsymbol{f}}\mathcal{L}_{\boldsymbol{x}}\|_2 \leq \gamma$ for all $\boldsymbol{x} \in \mathcal{X}$),

$$\gamma^{-1}\left\|\nabla_{\boldsymbol{\theta}_0(\mathcal{A})}\mathcal{L}_{\boldsymbol{x}}\right\|_2 = \gamma^{-1}\left\|\nabla_{\boldsymbol{f}}\mathcal{L}_{\boldsymbol{x}}^{\top}\nabla_{\boldsymbol{\theta}_0(\mathcal{A})}\boldsymbol{f}(\boldsymbol{x}, \boldsymbol{\theta}_0(\mathcal{A}))\right\|_2 \leq \left\|\nabla_{\boldsymbol{\theta}_0(\mathcal{A})}\boldsymbol{f}(\boldsymbol{x}, \boldsymbol{\theta}_0(\mathcal{A}))\right\|_{\text{F}}. \tag{10}$$

The Frobenius norm $\left\|\nabla_{\boldsymbol{\theta}_0(\mathcal{A})}\boldsymbol{f}(\boldsymbol{x}, \boldsymbol{\theta}_0(\mathcal{A}))\right\|_{\text{F}}$ can thus be approximated efficiently using its lower bound in (10) through automatic differentiation (Baydin et al., 2017).

Meanwhile, the evaluation of $\|\boldsymbol{\Theta}_0(\mathcal{A})\|_{\text{tr}}$ in (9) requires iterating over the entire dataset of size $m$, which thus incurs $\mathcal{O}(m)$ time. Fortunately, this incurred time can be reduced by parallelization over

---

[2]The same learning rate is shared among all candidate architectures for their model training during the search process in conventional NAS algorithms such as (Liu et al., 2019).

[3]Notably, the $\gamma$-Lipschitz continuity is well-satisfied by widely adopted loss functions (see Appendix A.3).

mini-batches. Let the set $\mathcal{X}_j$ denote the input feature vectors of the $j$-th randomly sampled mini-batch of size $|\mathcal{X}_j| = b$. By combining (9) and (10),

$$\|\mathbf{\Theta}_0(\mathcal{A})\|_{\mathrm{tr}} \geq \gamma^{-1} \sum_{\boldsymbol{x} \in \mathcal{X}} \left\|\nabla_{\boldsymbol{\theta}_0(\mathcal{A})} \mathcal{L}_{\boldsymbol{x}}\right\|_2^2 \geq b\gamma^{-1} \sum_{j=1}^{m/b} \left\|b^{-1} \sum_{\boldsymbol{x} \in \mathcal{X}_j} \nabla_{\boldsymbol{\theta}_0(\mathcal{A})} \mathcal{L}_{\boldsymbol{x}}\right\|_2^2, \quad (11)$$

where the last inequality follows from Jensen's inequality. Note that (11) provides an approximation of $\|\mathbf{\Theta}_0(\mathcal{A})\|_{\mathrm{tr}}$ incurring $\mathcal{O}(m/b)$ time because the gradients within a mini-batch can be evaluated in parallel. In practice, we further approximate the summation over $m/b$ mini-batches in (11) by using only one single uniformly randomly sampled mini-batch $\mathcal{X}_j$. Formally, under the definition of $\|\widetilde{\mathbf{\Theta}}_0(\mathcal{A})\|_{\mathrm{tr}} \triangleq \|b^{-1} \sum_{\boldsymbol{x} \in \mathcal{X}_j} \nabla_{\boldsymbol{\theta}_0(\mathcal{A})} \mathcal{L}_{\boldsymbol{x}}\|_2^2$, our final approximation of $\|\mathbf{\Theta}_0(\mathcal{A})\|_{\mathrm{tr}}$ becomes

$$\|\mathbf{\Theta}_0(\mathcal{A})\|_{\mathrm{tr}} \approx m\gamma^{-1}\|\widetilde{\mathbf{\Theta}}_0(\mathcal{A})\|_{\mathrm{tr}}. \quad (12)$$

This final approximation incurs only $\mathcal{O}(1)$ time and can also characterize the performance of neural architectures effectively, as demonstrated in our experiments. Interestingly, a similar form to (12), called gradient flow (Wang et al., 2020), has also been applied in network pruning at initialization.

### 3.3 Optimization and Search Algorithm

The approximation of $\|\mathbf{\Theta}_0(\mathcal{A})\|_{\mathrm{tr}}$ in Sec. 3.2 engenders an efficient optimization of our reformulated NAS in (8): Firstly, we apply a penalty method to transform (8) into an unconstrained optimization problem. Given a penalty coefficient $\mu$ and an exterior penalty function $F(x) \triangleq \max(0, x)$ with a pre-defined constant $\nu \triangleq \gamma n \eta^{-1}$, and a randomly sampled mini-batch $\mathcal{X}_j$, by replacing $\|\mathbf{\Theta}_0(\mathcal{A})\|_{\mathrm{tr}}$ with the approximation in (12), our reformulated NAS problem (8) can be transformed into

$$\max_{\mathcal{A}} \left[\|\widetilde{\mathbf{\Theta}}_0(\mathcal{A})\|_{\mathrm{tr}} - \mu F(\|\widetilde{\mathbf{\Theta}}_0(\mathcal{A})\|_{\mathrm{tr}} - \nu)\right]. \quad (13)$$

Interestingly, (13) implies that the complexity of the final selected architectures is limited by not only the constraint $\nu$ (discussed in Sec. 3.1) but also the penalty coefficient $\mu$: For a fixed constant $\nu$, a larger $\mu$ imposes a stricter limitation on the complexity of architectures (i.e., $\|\widetilde{\mathbf{\Theta}}_0(\mathcal{A})\|_{\mathrm{tr}} < \nu$) in the optimization of (13).

The optimization of (13) in the discrete search space, however, is intractable. So, we apply certain optimization tricks to simplify it: Following that of Liu et al. (2019); Xie et al. (2019), we represent the search space as a one-shot architecture such that candidate architectures in the search space can be represented as the sub-graphs of this one-shot architecture. Next, instead of optimizing (13), we introduce a distribution $p_{\boldsymbol{\alpha}}(\mathcal{A})$ (parameterized by $\boldsymbol{\alpha}$) over the candidate architectures in this search space like that in (Zoph & Le, 2017; Pham et al., 2018; Xie et al., 2019) to optimize the expected performance of architectures sampled from $p_{\boldsymbol{\alpha}}(\mathcal{A})$:

$$\max_{\boldsymbol{\alpha}} \mathbb{E}_{\mathcal{A} \sim p_{\boldsymbol{\alpha}}(\mathcal{A})} \left[R(\mathcal{A})\right] \quad \text{s.t. } R(\mathcal{A}) \triangleq \|\widetilde{\mathbf{\Theta}}_0(\mathcal{A})\|_{\mathrm{tr}} - \mu F(\|\widetilde{\mathbf{\Theta}}_0(\mathcal{A})\|_{\mathrm{tr}} - \nu). \quad (14)$$

Then, we apply Gumbel-Softmax (Jang et al., 2017; Maddison et al., 2017) to relax the optimization of (14) to be continuous and differentiable using the reparameterization trick. Specifically, for a given $\boldsymbol{\alpha}$, to sample an architecture $\mathcal{A}$, we simply need to sample $\boldsymbol{g}$ from $p(\boldsymbol{g}) = \mathrm{Gumbel}(\mathbf{0}, \mathbf{1})$ and then determine $\mathcal{A}$ using $\boldsymbol{\alpha}$ and $\boldsymbol{g}$ (more details in Appendix B.3). Consequently, (14) can be transformed into

$$\max_{\boldsymbol{\alpha}} \mathbb{E}_{\boldsymbol{g} \sim p(\boldsymbol{g})} \left[R(\mathcal{A}(\boldsymbol{\alpha}, \boldsymbol{g}))\right]. \quad (15)$$

After that, we approximate (15) based on its first-order Taylor expansion at initialization such that it can be optimized efficiently through a gradient-based algorithm. In particular, given the first-order approximation within the $\xi$-neighborhood of initialization $\boldsymbol{\alpha}_0$ (i.e., $\|\Delta\|_2 \leq \xi$):

$$\mathbb{E}_{\boldsymbol{g} \sim p(\boldsymbol{g})} \left[R(\mathcal{A}(\boldsymbol{\alpha}_0 + \Delta, \boldsymbol{g}))\right] \approx \mathbb{E}_{\boldsymbol{g} \sim p(\boldsymbol{g})} \left[R(\mathcal{A}(\boldsymbol{\alpha}_0, \boldsymbol{g})) + \nabla_{\boldsymbol{\alpha}_0} R(\mathcal{A}(\boldsymbol{\alpha}_0, \boldsymbol{g}))^\top \Delta\right], \quad (16)$$

the maximum of (16) is achieved when

$$\Delta^* = \underset{\|\Delta\|_2 \leq \xi}{\arg\max} \, \mathbb{E}_{\boldsymbol{g} \sim p(\boldsymbol{g})} \left[\nabla_{\boldsymbol{\alpha}_0} R(\mathcal{A}(\boldsymbol{\alpha}_0, \boldsymbol{g}))^\top \Delta\right] = \xi \, \mathbb{E}_{\boldsymbol{g} \sim p(\boldsymbol{g})} \left[\frac{\nabla_{\boldsymbol{\alpha}_0} R(\mathcal{A}(\boldsymbol{\alpha}_0, \boldsymbol{g}))}{\left\|\mathbb{E}_{\boldsymbol{g} \sim p(\boldsymbol{g})}[\nabla_{\boldsymbol{\alpha}_0} R(\mathcal{A}(\boldsymbol{\alpha}_0, \boldsymbol{g}))]\right\|_2}\right]. \quad (17)$$

---

**Algorithm 1** NAS at Initialization (NASI)

1: **Input:** dataset $\mathcal{D} \triangleq (\mathcal{X}, \mathcal{Y})$, batch size $b$, steps $T$, penalty coefficient $\mu$, constraint constant $\nu$, initialized model parameters $\boldsymbol{\theta}_0$ for one-shot architecture and distribution $p_{\boldsymbol{\alpha}_0}(\mathcal{A})$ with initialization $\boldsymbol{\alpha}_0 = \mathbf{0}$, set $\xi = 1$
2: **for** step $t = 1, \ldots, T$ **do**
3:    Sample data $\mathcal{D}_t \sim \mathcal{D}$ of size $b$
4:    Sample $\boldsymbol{g}_t \sim p(\boldsymbol{g}) = \text{Gumbel}(\mathbf{0}, \mathbf{1})$ and determine sampled architecture $\mathcal{A}_t$ based on $\boldsymbol{\alpha}_0, \boldsymbol{g}_t$
5:    Evaluate gradient $G_t = \nabla_{\boldsymbol{\alpha}_0} R(\mathcal{A}_t)$ with data $\mathcal{D}_t$
6: **end for**
7: Estimate $\Delta^*$ with (18) and get $\boldsymbol{\alpha}^* = \boldsymbol{\alpha}_0 + \Delta^*$
8: Select architecture $\mathcal{A}^* = \arg\max_{\mathcal{A}} p_{\boldsymbol{\alpha}^*}(\mathcal{A})$

---

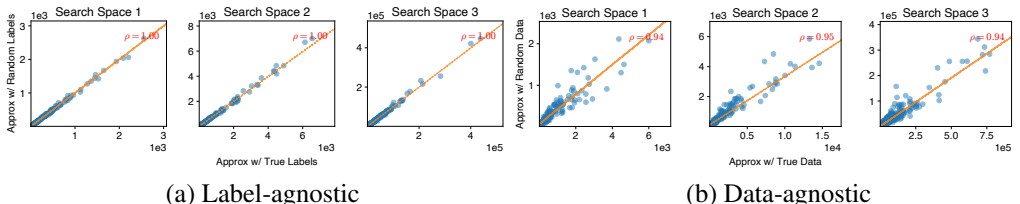

(a) Label-agnostic          (b) Data-agnostic

Figure 1: Comparison of the approximated $\|\boldsymbol{\Theta}_0(\mathcal{A})\|_{\text{tr}}$ following (12) using the three search spaces of NAS-Bench-1Shot1 on CIFAR-10 (a) between random vs. true labels, and (b) between random vs. true data. Each pair $(x, y)$ in the plots denotes the approximation for one candidate architecture in the search space with true vs. random labels (or data), respectively. The trends of these approximations are further illustrated by the lines in orange. In addition, Pearson correlation coefficient $\rho$ of the approximations with random vs. true labels (or data) is given in the corner.

The closed-form solution in (17) follows from the definition of dual norm and requires only a one-step optimization, i.e., without the iterative update of $\Delta$. Similar one-step optimizations have also been adopted in (Goodfellow et al., 2015; Wang et al., 2020).

Unfortunately, the expectation in (17) makes the evaluation of $\Delta^*$ intractable. Monte Carlo sampling is thus applied to estimate $\Delta^*$ efficiently: Given $T$ sequentially sampled $\boldsymbol{g}$ (i.e., $\boldsymbol{g}_1, \ldots, \boldsymbol{g}_T$) and let $G_i \triangleq \nabla_{\boldsymbol{\alpha}_0} R(\mathcal{A}(\boldsymbol{\alpha}_0, \boldsymbol{g}_i))$, $\Delta^*$ can be approximated as

$$\Delta^* \approx \frac{\xi}{T} \sum_{t=1}^{T} \frac{G_t}{\max(\|G_1\|_2, \ldots, \|G_t\|_2)} \ . \tag{18}$$

Note that inspired by AMSGrad (Reddi et al., 2018), the expectation $\left\|\mathbb{E}_{\boldsymbol{g} \sim p(\boldsymbol{g})}[\nabla_{\boldsymbol{\alpha}_0} R(\mathcal{A}(\boldsymbol{\alpha}_0, \boldsymbol{g}))]\right\|_2$ in (17) has been approximated using $\max(\|G_1\|_2, \ldots, \|G_t\|_2)$ in (18) based on a sample of $\boldsymbol{g}$ at time $t$. Interestingly, this approximation is non-decreasing in $t$ and therefore achieves a similar effect of learning rate decay, which may lead to a better-behaved optimization of $\Delta^*$. With the optimal $\Delta^*$ and $\boldsymbol{\alpha}^* = \boldsymbol{\alpha}_0 + \Delta^*$, the final architecture can then be selected as $\mathcal{A}^* \triangleq \arg\max_{\mathcal{A}} p_{\boldsymbol{\alpha}^*}(\mathcal{A})$, which finally completes our NAS at Initialization (NASI) algorithm detailed in Algorithm 1. Interestingly, this simple and efficient solution in (18) can already allow us to select architectures with competitive performances, as shown in our experiments (Sec. 5).

## 4  LABEL- AND DATA-AGNOSTIC SEARCH OF NASI

Besides the improved search efficiency by completely avoiding model training during search, NASI can even guarantee the transferability of its selected architectures with its provable label- and data-agnostic search under mild conditions in Sec. 4.1 and Sec. 4.2, respectively. That is, under this provable label- and data-agnostic search, the final selected architectures on a proxy dataset using NASI are also likely to be selected and hence guaranteed to perform well on the target datasets. So, the transferability of the architectures selected via such label- and data-agnostic search can be naturally guaranteed, which will also be validated in Sec. 5 empirically.

### 4.1 Label-Agnostic Search

Our reformulated NAS problem (8) has explicitly revealed that it can be optimized without the need of the labels from a dataset. Though our approximation of $\|\mathbf{\Theta}_0(\mathcal{A})\|_{\mathrm{tr}}$ in (12) seemingly depends on the labels of a dataset, (12) can, however, be derived using random labels. This is because the Lipschitz continuity assumption on the loss function required by (10), which is necessary for the derivation of (12), remains satisfied when random labels are used. So, the approximation in (12) (and hence our optimization objective (13) that is based on this approximation) is label-agnostic, which hence justifies the label-agnostic nature of NASI. Interestingly, NAS algorithm achieving a similar label-agnostic search has also been developed in (Liu et al., 2020), which further implies the reasonableness of such label-agnostic search.

The label-agnostic approximation of $\|\mathbf{\Theta}_0(\mathcal{A})\|_{\mathrm{tr}}$ is demonstrated in Fig. 1a using the three search spaces of NAS-Bench-1Shot1 with randomly selected labels. According to Fig. 1a, the large Pearson correlation coefficient (i.e., $\rho \approx 1$) implies a strong correlation between the approximations with random vs. true labels, which consequently validates the label-agnostic approximation of $\|\mathbf{\Theta}_0(\mathcal{A})\|_{\mathrm{tr}}$. Overall, these empirical observations have verified that the approximation of $\|\mathbf{\Theta}_0(\mathcal{A})\|_{\mathrm{tr}}$ and hence NASI based on the optimization over this approximation are label-agnostic, which will be further validated empirically in Sec. 5.3.

### 4.2 Data-Agnostic Search

Besides being label-agnostic, NASI is also guaranteed to be data-agnostic. To justify this, we prove in Proposition 2 (following from the notations in Sec. 3.1) below that $\|\mathbf{\Theta}_0(\mathcal{A})\|_{\mathrm{tr}}$ is data-agnostic under mild conditions.

**Proposition 2.** *Suppose that $\boldsymbol{x} \in \mathbb{R}^{n_0}$ and $\|\boldsymbol{x}\|_2 \leq 1$ for all $\boldsymbol{x} \in \mathcal{X}$ given a dataset $(\mathcal{X}, \mathcal{Y})$ of size $|\mathcal{X}| = m$, a given $L$-layer neural architecture $\mathcal{A}$ is randomly initialized, and the $\gamma$-Lipschitz continuous nonlinearity $\sigma$ satisfies $|\sigma(x)| \leq |x|$. Then, for any two data distributions $P(\boldsymbol{x})$ and $Q(\boldsymbol{x})$, denote $Z \triangleq \int \|P(\boldsymbol{x}) - Q(\boldsymbol{x})\| \, \mathrm{d}\boldsymbol{x}$, as $n_1, \ldots, n_{L-1} \to \infty$ sequentially,*

$$(mn)^{-1} \left| \|\mathbf{\Theta}_0(\mathcal{A}; P)\|_{\mathrm{tr}} - \|\mathbf{\Theta}_0(\mathcal{A}; Q)\|_{\mathrm{tr}} \right| \leq n_0^{-1} Z D(\gamma)$$

*with probability arbitrarily close to 1. $D(\gamma)$ is set to $L$ if $\gamma = 1$, and $(1 - \gamma^{2L})/(1 - \gamma^2)$ otherwise.*

Its proof is in Appendix A.4. Proposition 2 reveals that for any neural architecture $\mathcal{A}$, $\|\mathbf{\Theta}_0(\mathcal{A})\|_{\mathrm{tr}}$ is data-agnostic if either one of the following conditions is satisfied: (a) Different datasets achieve a small $Z$ or (b) the input dimension $n_0$ is large. Interestingly, these two conditions required by the data-agnostic $\|\mathbf{\Theta}_0(\mathcal{A})\|_{\mathrm{tr}}$ can be well-satisfied in practice. Firstly, we always have $Z < 2$ according to the property of probability distributions. Moreover, many real-world datasets are indeed of high dimensions such as $\sim 10^3$ for CIFAR-10 (Krizhevsky et al., 2009) and $\sim 10^5$ for COCO (Lin et al., 2014). Since $\|\mathbf{\Theta}_0(\mathcal{A})\|_{\mathrm{tr}}$ under such mild conditions is data-agnostic, NASI using $\|\mathbf{\Theta}_0(\mathcal{A})\|_{\mathrm{tr}}$ as the optimization objective in (8) is also data-agnostic.

While $\|\mathbf{\Theta}_0(\mathcal{A})\|_{\mathrm{tr}}$ is costly to evaluate, we demonstrate in Fig. 1b that the approximated $\|\mathbf{\Theta}_0(\mathcal{A})\|_{\mathrm{tr}}$ in (12) is also data-agnostic by using random data generated with the standard Gaussian distribution. Similar to the results using true vs. random labels in Fig. 1a, the approximated $\|\mathbf{\Theta}_0(\mathcal{A})\|_{\mathrm{tr}}$ using random vs. true data are also highly correlated by achieving a large Pearson correlation coefficient (i.e., $\rho > 0.9$). Interestingly, the correlation coefficient here is slightly smaller than the one of label-agnostic approximations in Fig. 1a, which implies that our approximated $\|\mathbf{\Theta}_0(\mathcal{A})\|_{\mathrm{tr}}$ is more agnostic to the labels than data. Based on these results, the approximated $\|\mathbf{\Theta}_0(\mathcal{A})\|_{\mathrm{tr}}$ is guaranteed to be data-agnostic. So, NASI based on the optimization over such a data-agnostic approximation is also data-agnostic, which will be further validated empirically in Sec. 5.3.

## 5 Experiments

### 5.1 Search in NAS-Bench-1Shot1

We firstly validate the search efficiency and effectiveness of our NASI in the three search spaces of NAS-Bench-1Shot1 (Zela et al., 2020b) on CIFAR-10. As these three search spaces are relatively

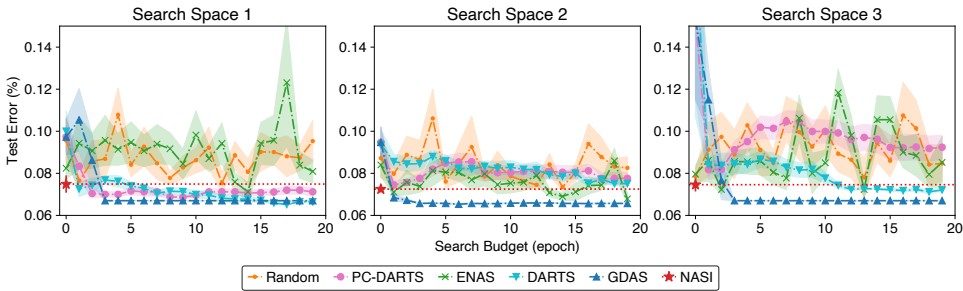

Figure 2: Comparison of search efficiency (search budget in $x$-axis) and effectiveness (test error evaluated on CIFAR-10 in $y$-axis) between NASI and other NAS algorithms in the three search spaces of NAS-Bench-1Shot1. The test error for each algorithm is reported with the mean and standard error after ten independent searches.

Table 1: Performance comparison among state-of-the-art (SOTA) image classifiers on CIFAR-10/100. The performance of the final architectures selected by NASI is reported with the mean and standard deviation of five independent evaluations. The search costs are evaluated on a single Nvidia 1080Ti.

| Architecture | Test Error (%) | | Params (M) | | Search Cost (GPU Hours) | Search Method |
|---|---|---|---|---|---|---|
| | C10 | C100 | C10 | C100 | | |
| DenseNet-BC (Huang et al., 2017) | 3.46* | 17.18* | 25.6 | 25.6 | - | manual |
| NASNet-A (Zoph et al., 2018) | 2.65 | - | 3.3 | - | 48000 | RL |
| AmoebaNet-A (Real et al., 2019) | 3.34±0.06 | 18.93† | 3.2 | 3.1 | 75600 | evolution |
| PNAS (Liu et al., 2018) | 3.41±0.09 | 19.53* | 3.2 | 3.2 | 5400 | SMBO |
| ENAS (Pham et al., 2018) | 2.89 | 19.43* | 4.6 | 4.6 | 12 | RL |
| NAONet (Luo et al., 2018) | 3.53 | - | 3.1 | - | 9.6 | NAO |
| DARTS (2nd) (Liu et al., 2019) | 2.76±0.09 | 17.54† | 3.3 | 3.4 | 24 | gradient |
| GDAS (Dong & Yang, 2019) | 2.93 | 18.38 | 3.4 | 3.4 | 7.2 | gradient |
| NASP (Yao et al., 2020) | 2.83±0.09 | - | 3.3 | - | 2.4 | gradient |
| P-DARTS (Chen et al., 2019) | 2.50 | - | 3.4 | - | 7.2 | gradient |
| DARTS- (avg) (Chu et al., 2020) | 2.59±0.08 | 17.51±0.25 | 3.5 | 3.3 | 9.6 | gradient |
| SDARTS-ADV (Chen & Hsieh, 2020) | 2.61±0.02 | - | 3.3 | - | 31.2 | gradient |
| R-DARTS (L2) (Zela et al., 2020a) | 2.95±0.21 | 18.01±0.26 | - | - | 38.4 | gradient |
| TE-NAS♯ (Chen et al., 2021) | 2.83±0.06 | 17.42±0.56 | 3.8 | 3.9 | 1.2 | training-free |
| NASI-FIX | 2.79±0.07 | 16.12±0.38 | 3.9 | 4.0 | **0.24** | training-free |
| NASI-ADA | 2.90±0.13 | 16.84±0.40 | 3.7 | 3.8 | **0.24** | training-free |

† Reported by Dong & Yang (2019) with their experimental settings.
* Obtained by training corresponding architectures without cutout (Devries & Taylor, 2017) augmentation.
♯ Evaluated using our experimental settings in Appendix B.4.

small, a lower penalty coefficient $\mu$ and a larger constraint $\nu$ (i.e., $\mu=1$ and $\nu=1000$) are adopted to encourage the selection of high-complexity architectures in the optimization of (13). Here, $\nu$ is determined adaptively as shown in Appendix B.1.

Figure 2 shows the results comparing the efficiency and effectiveness between NASI with a one-epoch search budget and other NAS algorithms with a maximum search budget of 20 epochs to allow sufficient model training during their search process. Figure 2 reveals that among all these three search spaces, NASI consistently selects architectures of better generalization performance than other NAS algorithms with a search budget of only one epoch. Impressively, the architectures selected by our one-epoch NASI can also achieve performances that are comparable to the best-performing NAS algorithms with $19\times$ more search budget. Above all, NASI guarantees its benefits of improving the search efficiency of NAS algorithms considerably without sacrificing the generalization performance of its selected architectures.

## 5.2 SEARCH IN THE DARTS SEARCH SPACE

We then compare NASI with other NAS algorithms in a more complex search space than NAS-Bench-1Shot1, i.e., the DARTS (Liu et al., 2019) search space (detailed in Appendix B.2). Here, NASI

Table 2: Performance comparison of the architectures selected by NASI with random vs. true labels/data on CIFAR-10. The standard method denotes the search with the true labels and data of CIFAR-10 and each test error in the table is reported with the mean and standard deviation of five independent searches.

| Method | NAS-Bench-1Shot1 | | | DARTS |
|---|---|---|---|---|
| | S1 | S2 | S3 | |
| Standard | 7.3±1.1 | 7.2±0.4 | 7.2±0.6 | 2.95±0.13 |
| Random Label | 6.8±0.3 | 7.0±0.4 | 7.5±1.4 | 2.90±0.12 |
| Random Data | 6.6±0.2 | 7.5±0.7 | 7.3±0.9 | 2.97±0.10 |

selects the architecture with a search budget of $T=100$, batch size of $b=64$ and $\mu=2$. Besides, two different methods are applied to determine the constraint $\nu$ during the search process: the adaptive determination with an initial value of 500 and the fixed determination with a value of 100. The final selected architectures with adaptive and fixed $\nu$ are, respectively, called NASI-ADA and NASI-FIX (visualized in Appendix C.4), which are then evaluated on CIFAR-10/100 (Krizhevsky et al., 2009) and ImageNet (Deng et al., 2009) following Appendix B.4.

Table 1 summarizes the generalization performance of the final architectures selected by various NAS algorithms on CIFAR-10/100. Compared with popular training-based NAS algorithms, NASI achieves a substantial improvement in search efficiency and maintains a competitive generalization performance. Even when compared with other training-free NAS algorithm (i.e., TE-NAS), NASI is also able to select competitive or even outperformed architectures with a smaller search cost. Besides, NASI-FIX achieves the smallest test error on CIFAR-100, which also demonstrates the transferability of the architectures selected by NASI over different datasets. Consistent results on ImageNet can be found in Appendix C.5.

### 5.3 LABEL- AND DATA-AGNOSTIC SEARCH

To further validate the label- and data-agnostic search achieved by our NASI as discussed in Sec. 4, we compare the generalization performance of the final architectures selected by NASI using random labels and data on CIFAR-10. The random labels are randomly selected from all possible categories while the random data is i.i.d. sampled from the standard Gaussian distribution. Both NAS-Bench-1Shot1 and the DARTS search space are applied in this performance comparison where the same search and training settings in Sec. 5.1 and Sec. 5.2 are adopted.

Table 2 summarizes the performance comparison. Interestingly, among all these four search spaces, comparable generalization performances are obtained on CIFAR-10 for both the architectures selected with random labels (or data) and the ones selected with true labels and data. These results hence confirm the label- and data-agnostic search achieved by NASI, which therefore also further validates the transferability of the architectures selected by NASI over different datasets.

## 6 CONCLUSION

This paper describes a novel NAS algorithm called NASI that exploits the capability of NTK for estimating the performance of candidate architectures at initialization. Consequently, NASI can completely avoid model training during the search process to achieve higher search efficiency than existing NAS algorithms. NASI can also achieve competitive generalization performance across different search spaces and benchmark datasets. Interestingly, NASI is guaranteed to be label- and data-agnostic under mild conditions, which therefore implies the transferability of the final selected architectures by NASI over different datasets. With all these advantages, NASI can thus be adopted to select well-performing architectures for unsupervised tasks and larger-scale datasets efficiently, which to date remains challenging to other training-based NAS algorithms. Furthermore, NASI can also be integrated into other training-based one-shot NAS algorithms to improve their search efficiency while preserving the search effectiveness of these training-based algorithms.

## Reproducibility Statement

To guarantee the reproducibility of the theoretical analysis in this paper, we have provided complete proof of our propositions and also the justification of certain assumptions in Appendix A. Moreover, we have conducted adequate ablation studies to further investigate the impacts of these assumptions as well as the approximations used in our method on the final search results in Appendix C.6. Meanwhile, to guarantee the reproducibility of the empirical results in this paper, we have provided our codes in the supplementary materials and detailed experimental settings in Appendix B.

## Acknowledgments

This research is supported by A*STAR under its RIE2020 Advanced Manufacturing and Engineering (AME) Programmatic Funds (Award A20H6b0151).

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

## APPENDIX A  THEOREMS AND PROOFS

We firstly introduce the theorems proposed by Jacot et al. (2018); Lee et al. (2019a), which reveal the capability of NTK in characterizing the training dynamics of infinite-width DNNs. Note that the these theorems follow the parameterization and initialization of DNNs in (Jacot et al., 2018). Our analysis based on these theorems thus also needs to follow such parameterization and initialization of DNNs.

### A.1  NEURAL TANGENT KERNEL AT INITIALIZATION

Jacot et al. (2018) have validated that the outputs of infinite-width DNNs at initialization tends to Gaussian process (Theorem 1) and have further revealed the deterministic limit of NTK at initialization (Theorem 2). We denote $\mathbb{I}_{n_L}$ as a $n_L \times n_L$ matrix with all elements being 1.

**Theorem 1** (Jacot et al. (2018)). *For a network of depth $L$ at initialization, with a Lipschitz nonlinearity $\sigma$, and in the limit as $n_1, \cdots, n_{L-1} \to \infty$ sequentially, the output functions $f_{\boldsymbol{\theta},i}$ for $i = 1, \cdots, n_L$, tend (in law) to iid centered Gaussian processes of covariance $\Sigma^{(L)}$, where $\Sigma^{(L)}$ is defined recursively by*

$$\Sigma^{(1)}(\boldsymbol{x}, \boldsymbol{x}') = \frac{1}{n_0} \boldsymbol{x}^\top \boldsymbol{x}' + \beta^2 \, ,$$

$$\Sigma^{(L)}(\boldsymbol{x}, \boldsymbol{x}') = \mathbb{E}_{g \sim \mathcal{N}(0, \Sigma^{(L-1)})} \left[ \sigma(g(\boldsymbol{x})) \sigma(g(\boldsymbol{x}')) \right] + \beta^2$$

*such that the expectation is with respect to a centered Gaussian process $\boldsymbol{f}$ with covariance $\Sigma^{(L-1)}$.*

**Theorem 2** (Jacot et al. (2018)). *For a network of depth $L$ at initialization, with a Lipschitz nonlinearity $\sigma$, and in the limit as $n_1, \cdots, n_{L-1} \to \infty$ sequentially, the NTK $\boldsymbol{\Theta}^{(L)}$ converges in probability to a deterministic limiting kernel:*

$$\boldsymbol{\Theta}^{(L)} \to \boldsymbol{\Theta}_\infty^{(L)} \otimes \mathbb{I}_{n_L} \, .$$

*Kernel $\boldsymbol{\Theta}_\infty^{(L)} : \mathbb{R}^{n_0 \times n_0} \to \mathbb{R}$ is defined recursively by*

$$\boldsymbol{\Theta}_\infty^{(1)}(\boldsymbol{x}, \boldsymbol{x}') = \Sigma^{(1)}(\boldsymbol{x}, \boldsymbol{x}') \, ,$$

$$\boldsymbol{\Theta}_\infty^{(L)}(\boldsymbol{x}, \boldsymbol{x}') = \boldsymbol{\Theta}_\infty^{(L-1)}(\boldsymbol{x}, \boldsymbol{x}') \dot{\Sigma}^{(L)}(\boldsymbol{x}, \boldsymbol{x}') + \Sigma^{(L)}(\boldsymbol{x}, \boldsymbol{x}') \, ,$$

*where*

$$\dot{\Sigma}^{(L)}(\boldsymbol{x}, \boldsymbol{x}') = \mathbb{E}_{g \sim \mathcal{N}(0, \Sigma^{(L-1)})} \left[ \dot{\sigma}(g(\boldsymbol{x})) \dot{\sigma}(g(\boldsymbol{x}')) \right]$$

*such that the expectation is with respect to a centered Gaussian process $\boldsymbol{f}$ with covariance $\Sigma^{(L-1)}$ and $\dot{\sigma}$ denotes the derivative of $\sigma$.*

### A.2  TRAINING DYNAMICS OF INFINITE-WIDTH NEURAL NETWORKS

Given $\lambda_{\min}(\boldsymbol{\Theta})$ as the minimal eigenvalue of NTK $\boldsymbol{\Theta}$ and define $\eta_{\text{critical}} \triangleq 2(\lambda_{\min}(\boldsymbol{\Theta}) + \lambda_{\max}(\boldsymbol{\Theta}))^{-1}$, Lee et al. (2019a) have characterized the training dynamics of infinite-wide neural networks as below.

**Theorem 3** (Lee et al. (2019a)). *Let $n_1 = \cdots = n_{L-1} = k$ and assume $\lambda_{\min}(\boldsymbol{\Theta}) > 0$. Applying gradient descent with learning rate $\eta < \eta_{\text{critical}}$ (or gradient flow), for every $\boldsymbol{x} \in \mathbb{R}^{n_0}$ with $\|\boldsymbol{x}\| \le 1$, with probability arbitrarily close to 1 over random initialization,*

$$\sup_{t \ge 0} \left\| \boldsymbol{f}_t - \boldsymbol{f}_t^{\text{lin}} \right\|_2 = \mathcal{O}(k^{-\frac{1}{2}}) \quad \text{as } k \to \infty \, .$$

**Remark.** For the case of $L = 2$, Du et al. (2019) have revealed that if any two input vectors of a dataset are not parallel, then $\lambda_{\min}(\boldsymbol{\Theta}) > 0$ holds, which fortunately can be well-satisfied for most real-world datasets. Though the training dynamics of DNNs only tend to be governed by their linearization at initialization when the infinite width is satisfied as revealed in Theorem 3, empirical results in (Lee et al., 2019a) suggest that such linearization can also govern the training dynamics of practical over-parameterized DNNs.

A.3 PROOF OF PROPOSITION 1

With aforementioned theorems, especially Theorem 3, our Proposition 1 can be proved as below with an introduced lemma (Lemma 1).

**Lemma 1.** *Let loss function $\mathcal{L}\left(\boldsymbol{f}(\mathcal{X};\boldsymbol{\theta}_t),\mathcal{Y}\right)$, abbreviated to $\mathcal{L}(\boldsymbol{f}_t)$, be $\gamma$-Lipschitz continuous within the domain $\mathcal{V}$. Under the condition in Theorem 3, there exists a constant $c_0 > 0$ such that as $k \to \infty$,*

$$\left\|\mathcal{L}(\boldsymbol{f}_t) - \mathcal{L}(\boldsymbol{f}_t^{\text{lin}})\right\|_2 \leq \frac{c_0\gamma}{\sqrt{k}}$$

*with probability arbitrarily close to 1.*

*Proof.* Since $\mathcal{L}(\boldsymbol{f}_t)$ is $\gamma$-Lipschitz continuous, for any $\boldsymbol{v}, \boldsymbol{u} \in \mathcal{V}$, we have

$$\|\mathcal{L}(\boldsymbol{v}) - \mathcal{L}(\boldsymbol{u})\|_2 \leq \gamma \|\boldsymbol{v} - \boldsymbol{u}\|_2 \tag{19}$$

following the definition of Lipschitz continuity. Besides, under the condition in Theorem 3, Theorem 3 reveals that there exists a constant $c_0$ such that as $k \to \infty$,

$$\left\|\boldsymbol{f}_t - \boldsymbol{f}_t^{\text{lin}}\right\|_2 \leq \frac{c_0}{\sqrt{k}} \tag{20}$$

with probability arbitrarily close to 1. Combining (19) and (20), we hence can finish the proof by

$$\left\|\mathcal{L}(\boldsymbol{f}_t) - \mathcal{L}(\boldsymbol{f}_t^{\text{lin}})\right\|_2 \leq \gamma \left\|\boldsymbol{f}_t - \boldsymbol{f}_t^{\text{lin}}\right\|_2 \leq \frac{c_0\gamma}{\sqrt{k}} \quad \text{as } k \to \infty . \tag{21}$$

$\square$

**Remark.** The $\gamma$-Lipschitz continuity based on $\|\cdot\|_2$ is commonly satisfied for widely adopted loss functions. For example, given 1-dimensional MSE $\mathcal{L} = m^{-1}\sum_i^m (x_i - y_i)^2$, let $x_i, y_i \in [0,1]$ denote the prediction and label respectively, the Lipschitz of MSE with respect to $\boldsymbol{x} \triangleq (x_1, \cdots, x_m)^\top$ is 2. Meanwhile, given the $n$-class Cross Entropy with Softmax $\mathcal{L} = -m^{-1}\sum_i^m \sum_j^n y_{i,j}\log(p_{i,j})$ as the loss function, let $p_{i,j} \in (0,1)$ and $y_{i,j} \in \{0,1\}$ denote the prediction and label correspondingly, with $\sum_j y_{i,j} = 1$ and $p_{i,j} \triangleq \exp(x_{i,j})/\sum_j \exp(x_{i,j})$ for input $x_{i,j} \in \mathbb{R}$, the Lipschitz of Cross Entropy with Softmax with respect to $\boldsymbol{x} \triangleq (x_{1,1}, \cdots, x_{i,j}, \cdots, x_{m,n})^\top$ is then 1.

**Proof of Proposition 1.** Note that the linearization $\boldsymbol{f}^{\text{lin}}(\boldsymbol{x};\boldsymbol{\theta}_t)$ achieves a constant NTK $\boldsymbol{\Theta}_t = \boldsymbol{\Theta}_0$ because its gradient with respect to $\boldsymbol{\theta}_t$ (i.e., $\nabla_{\boldsymbol{\theta}_0}\boldsymbol{f}(\boldsymbol{x};\boldsymbol{\theta}_0)$) stays constant over time. Given MSE as the loss function, according to the loss decomposition (4) in Sec. 2.2, the training dynamics of $\boldsymbol{f}^{\text{lin}}(\boldsymbol{x};\boldsymbol{\theta}_t)$ can then be analyzed in a closed form:

$$\mathcal{L}(\boldsymbol{f}_t^{\text{lin}}) = \frac{1}{m}\sum_{i=1}^{mn}(1 - \eta\lambda_i)^{2t}(\boldsymbol{u}_i^\top \mathcal{Y})^2 , \tag{22}$$

where $\boldsymbol{\Theta}_0 = \sum_{i=1}^{mn}\lambda_i(\boldsymbol{\Theta}_0)\boldsymbol{u}_i\boldsymbol{u}_i^\top$, and $\lambda_i(\boldsymbol{\Theta}_0)$ and $\boldsymbol{u}_i$ denote the $i$-th largest eigenvalue and the corresponding eigenvector of $\boldsymbol{\Theta}_0$, respectively. With $\eta < \lambda_{\max}(\boldsymbol{\Theta}_0)^{-1}$ and $\lambda_{\min}(\boldsymbol{\Theta}_0) > 0$, $\eta\lambda_i(\boldsymbol{\Theta}_0)$ is then under the constraint that

$$0 < \eta\lambda_i(\boldsymbol{\Theta}_0) < \frac{\lambda_{\max}(\boldsymbol{\Theta}_0)}{\lambda_{\max}(\boldsymbol{\Theta}_0)} \Rightarrow 0 < \eta\lambda_i(\boldsymbol{\Theta}_0) < 1 . \tag{23}$$

Hence, for the case of $t \geq 0.5$, with $0 < 1 - \eta\lambda_i(\boldsymbol{\Theta}_0) < 1$ and $\overline{\lambda}(\boldsymbol{\Theta}_0) \triangleq (mn)^{-1}\sum_{i=1}^{mn}\lambda_i(\boldsymbol{\Theta}_0)$,

$$\sum_{i=1}^{mn}(1 - \eta\lambda_i(\boldsymbol{\Theta}_0))^{2t} \leq \sum_{i=1}^{mn}(1 - \eta\lambda_i(\boldsymbol{\Theta}_0))$$
$$= mn(1 - \eta\overline{\lambda}(\boldsymbol{\Theta}_0)) . \tag{24}$$

Further, given $0 \leq t < 0.5$, the scalar function $y = x^{2t}$ is concave for any $x \in \mathbb{R}_{\geq 0}$. Following from the Jensen's inequality on this concave function,

$$\sum_{i=1}^{mn}(1 - \eta\lambda_i(\boldsymbol{\Theta}_0))^{2t} \leq mn\left[\frac{1}{mn}\sum_{i=1}^{mn}(1 - \eta\lambda_i(\boldsymbol{\Theta}_0))\right]^{2t}$$
$$= mn(1 - \eta\overline{\lambda}(\boldsymbol{\Theta}_0))^{2t} . \tag{25}$$

With bounded labels $\mathcal{Y} \in [0, 1]^{mn}$ and the unit norm of eigenvectors $\|\boldsymbol{u}_i\|_2 = 1$,

$$(\boldsymbol{u}_i^\top \mathcal{Y})^2 \leq \|\boldsymbol{u}_i\|_2^2 \|\mathcal{Y}\|_2^2 \leq \|\mathcal{Y}\|_2^2 \leq mn \,. \tag{26}$$

By introducing (24), (25) and (26) into (22), the training loss $\mathcal{L}(\boldsymbol{f}_t^{\mathrm{lin}})$ can then be bounded by

$$\begin{aligned}
\mathcal{L}(\boldsymbol{f}_t^{\mathrm{lin}}) &\leq (mn) \cdot \frac{1}{m} \sum_{i=1}^{mn} (1 - \lambda_i(\boldsymbol{\Theta}_0))^{2t} \\
&\leq mn^2 (1 - \eta \bar{\lambda}(\boldsymbol{\Theta}_0))^q \,,
\end{aligned} \tag{27}$$

where $q$ is set to be $2t$ if $0 \leq t < 0.5$, and 1 otherwise.

Following from Theorem 3, while applying gradient descent (or gradient flow) with learning rate $\eta < \lambda_{\max}^{-1} < \eta_{\mathrm{critical}} \triangleq 2(\lambda_{\min} + \lambda_{\max})^{-1}$, for all $\boldsymbol{x} \in \mathcal{X}$ with $\|\boldsymbol{x}\| \leq 1$, there exists a constant $c_0$ such that as $k \to \infty$,

$$\left\| \boldsymbol{f}_t - \boldsymbol{f}_t^{\mathrm{lin}} \right\|_2 \leq \frac{c_0}{\sqrt{k}} \tag{28}$$

with probability arbitrarily close to 1. Hence, $\boldsymbol{f}_t^{\mathrm{lin}} \in \left[ -c_0 \sqrt{1/k}, 1 + c_0 \sqrt{1/k} \right]^{mn}$ with $\boldsymbol{f}_t \in [0, 1]^{mn}$. Within the extended domain $\boldsymbol{f}_t \in \left[ -c_0 \sqrt{1/k}, 1 + c_0 \sqrt{1/k} \right]^{mn}$ for the MSE loss function,

$$\begin{aligned}
\|\nabla_{\boldsymbol{f}_t} \mathcal{L}\|_2 &= \frac{2}{m} \|\mathcal{Y} - \boldsymbol{f}_t\|_2 \\
&\leq 2\sqrt{n/m} \left( 1 + c_0 \sqrt{1/k} \right) \,.
\end{aligned} \tag{29}$$

Hence, the MSE within this extended domain is $2\sqrt{n/m} \left( 1 + c_0 \sqrt{1/k} \right)$-Lipschitz continuous.

Combining with Lemma 1,

$$\begin{aligned}
\mathcal{L}(\boldsymbol{f}_t) &\leq \mathcal{L}(\boldsymbol{f}_t^{\mathrm{lin}}) + 2c_0 \sqrt{n/(mk)} \left( 1 + c_0 \sqrt{1/k} \right) \\
&\leq mn^2 (1 - \eta \bar{\lambda})^q + 2c_0 \sqrt{n/(mk)} \left( 1 + c_0 \sqrt{1/k} \right) \,,
\end{aligned} \tag{30}$$

which concludes the proof by denoting $\mathcal{L}_t \triangleq \mathcal{L}(\boldsymbol{f}_t)$.

**Remark.** $\|\boldsymbol{x}\|_2 \leq 1$ can be well-satisfied for the normalized dataset, which is conventionally adopted as data pre-processing in practice.

### A.4 PROOF OF PROPOSITION 2

According to Theorem 2, $\boldsymbol{\Theta}_0 \triangleq \boldsymbol{\Theta}^{(L)}$ is determined by both $\Sigma^{(L)}(\boldsymbol{x}, \boldsymbol{x})$ and $\dot{\Sigma}^{(L)}(\boldsymbol{x}, \boldsymbol{x})$ of all $\boldsymbol{x} \in \mathcal{X}$. We hence introduce Lemma 2 below regarding $\Sigma^{(L)}(\boldsymbol{x}, \boldsymbol{x})$ and $\dot{\Sigma}^{(L)}(\boldsymbol{x}, \boldsymbol{x})$ to ease the proof of Proposition 2. Particularly, we set $\beta = 0$ for the $\beta$ in Theorem 1 and Theorem 2 throughout our analysis. Note that this condition is usually satisfied by training DNNs without bias.

**Lemma 2.** *For a network of depth $L$ at initialization, with the $\gamma$-Lipschitz continuous nonlinearity $\sigma$ satisfying $|\sigma(x)| \leq |x|$ for all $x \in \mathbb{R}$, given the input features $\boldsymbol{x} \in \mathcal{X}$,*

$$\Sigma^{(L)}(\boldsymbol{x}, \boldsymbol{x}) \leq n_0^{-1} \boldsymbol{x}^\top \boldsymbol{x} \,, \quad \dot{\Sigma}^{(L)}(\boldsymbol{x}, \boldsymbol{x}) \leq \gamma^2 \,,$$

*where $\Sigma^{(L)}(\boldsymbol{x}, \boldsymbol{x})$ and $\dot{\Sigma}^{(L)}(\boldsymbol{x}, \boldsymbol{x})$ are defined in Theorem 1 and Theorem 2, respectively, and $\beta = 0$.*

*Proof.* Following from the definition in Theorem 1,

$$\begin{aligned}
\Sigma^{(L)}(\boldsymbol{x}, \boldsymbol{x}) &= \mathbb{E}_{g \sim \mathcal{N}(0, \Sigma^{(L-1)})} \left[ \sigma(g(\boldsymbol{x})) \sigma(g(\boldsymbol{x})) \right] \\
&\overset{(a)}{\leq} \mathbb{E}_{g \sim \mathcal{N}(0, \Sigma^{(L-1)})} \left[ g^2(\boldsymbol{x}) \right] \\
&\overset{(b)}{=} \Sigma^{(L-1)}(\boldsymbol{x}, \boldsymbol{x}) \,,
\end{aligned} \tag{31}$$

in which (a) follows from $|\sigma(x)| \leq |x|$ and (b) results from the variance of $g \sim \mathcal{N}(0, \Sigma^{(L-1)})$. By following (31) from layer $L$ to 1, we get

$$\Sigma^{(L)}(\boldsymbol{x}, \boldsymbol{x}) \leq \Sigma^{(1)}(\boldsymbol{x}, \boldsymbol{x}) \overset{(a)}{=} n_0^{-1} \boldsymbol{x}^\top \boldsymbol{x} , \tag{32}$$

where (a) is defined in Theorem 1.

Similarly, following from the definition in Theorem 2,

$$\begin{aligned}
\dot{\Sigma}^{(L)}(\boldsymbol{x}, \boldsymbol{x}) &= \mathbb{E}_{g \sim \mathcal{N}(0, \Sigma^{(L-1)})} [\dot{\sigma}(g(\boldsymbol{x}))\dot{\sigma}(g(\boldsymbol{x}))] \\
&\overset{(a)}{\leq} \mathbb{E}_{g \sim \mathcal{N}(0, \Sigma^{(L-1)})} [\gamma^2] \\
&\overset{(b)}{=} \gamma^2 ,
\end{aligned} \tag{33}$$

in which (a) results from the $\gamma$-Lipschitz continuity of nonlinearity and (b) follows from the expectation of a constant. $\qquad\square$

**Proof of Proposition 2.** Notably, Theorem 2 reveals that in the limit as $n_1, \cdots, n_{L-1} \to \infty$ sequentially, $\boldsymbol{\Theta}^{(L)} \to \boldsymbol{\Theta}_\infty^{(L)} \otimes \mathbb{I}_{n_L}$ with probability arbitrarily close to 1 over random initializations. We therefore only need to focus on this deterministic limiting kernel to simplify our analysis, i.e., let $\boldsymbol{\Theta}_0 = \boldsymbol{\Theta}_\infty^{(L)} \otimes \mathbb{I}_{n_L}$. Particularly, given $m$ input vectors $\boldsymbol{x} \sim P(\boldsymbol{x})$ with covariance matrix $\Sigma_P$ and a $L$-layer neural architecture of $n$-dimensional output, For $\gamma \neq 1$, we have

$$\begin{aligned}
(mn)^{-1} \|\boldsymbol{\Theta}_0\|_{\mathrm{tr}} &= (mn)^{-1} \|\boldsymbol{\Theta}_\infty^{(L)} \otimes \mathbb{I}_{n_L}\|_{\mathrm{tr}} \\
&\overset{(a)}{=} (mn)^{-1} \|\boldsymbol{\Theta}_\infty^{(L)}\|_{\mathrm{tr}} \|\mathbb{I}_{n_L}\|_{\mathrm{tr}} \\
&\overset{(b)}{=} m^{-1} \|\boldsymbol{\Theta}_\infty^{(L)}\|_{\mathrm{tr}} \\
&\overset{(c)}{=} \mathbb{E}_{\boldsymbol{x} \sim P(\boldsymbol{x})} \left[ \boldsymbol{\Theta}_\infty^{(L)}(\boldsymbol{x}, \boldsymbol{x}) \right] \\
&\overset{(d)}{=} \mathbb{E}_{\boldsymbol{x} \sim P(\boldsymbol{x})} \left[ \boldsymbol{\Theta}_\infty^{(L-1)}(\boldsymbol{x}, \boldsymbol{x}) \dot{\Sigma}^{(L)}(\boldsymbol{x}, \boldsymbol{x}) + \Sigma^{(L)}(\boldsymbol{x}, \boldsymbol{x}) \right] \\
&\overset{(e)}{\leq} \mathbb{E}_{\boldsymbol{x} \sim P(\boldsymbol{x})} \left[ \gamma^2 \boldsymbol{\Theta}_\infty^{(L-1)}(\boldsymbol{x}, \boldsymbol{x}) + n_0^{-1} \boldsymbol{x}^\top \boldsymbol{x} \right] \\
&\overset{(f)}{=} \mathbb{E}_{\boldsymbol{x} \sim P(\boldsymbol{x})} \left[ \gamma^2 \boldsymbol{\Theta}_\infty^{(L-1)}(\boldsymbol{x}, \boldsymbol{x}) \right] + n_0^{-1} \|\Sigma_P\|_{\mathrm{tr}} ,
\end{aligned} \tag{34}$$

in which (a) derives from the property of Kronecker product, (b) follows from the notation $n_L = n$ and (c) results from the property of expectation and trace norm. In addition, (d) follows from the definition of $\boldsymbol{\Theta}_\infty^{(L)}(\boldsymbol{x}, \boldsymbol{x})$ in Theorem 2 and (e) is derived by introducing Lemma 2 into (d). Finally, (f) follows from the expectation and variance of $P(\boldsymbol{x})$. By following (c-f) from layer $L$ to 1, we can get

$$\begin{aligned}
(mn)^{-1} \|\boldsymbol{\Theta}_0\|_{\mathrm{tr}} &\leq \gamma^{2(L-1)} \mathbb{E}_{\boldsymbol{x} \sim P(\boldsymbol{x})} \left[ \Sigma^{(1)}(\boldsymbol{x}, \boldsymbol{x}) \right] + \\
&\qquad n_0^{-1} \|\Sigma_P\|_{\mathrm{tr}} \left( 1 - \gamma^{2(L-1)} \right) \left( 1 - \gamma^2 \right)^{-1} \\
&= n_0^{-1} \|\Sigma_P\|_{\mathrm{tr}} \left( 1 - \gamma^{2L} \right) \left( 1 - \gamma^2 \right)^{-1} .
\end{aligned} \tag{35}$$

Notably, (35) is derived under the condition that $\gamma \neq 1$. For the case of $\gamma = 1$, similarly, we can get

$$(mn)^{-1} \|\boldsymbol{\Theta}_0\|_{\mathrm{tr}} \leq n_0^{-1} L \|\Sigma_P\|_{\mathrm{tr}} . \tag{36}$$

Note that with $\|\boldsymbol{x}\|_2 \leq 1$, we have

$$\begin{aligned}
\|\Sigma_P\|_{\mathrm{tr}} &= \mathbb{E}_{\boldsymbol{x} \sim P(\boldsymbol{x})} \left[ \boldsymbol{x}^\top \boldsymbol{x} \right] - \mathbb{E}_{\boldsymbol{x} \sim P(\boldsymbol{x})} \left[ \boldsymbol{x} \right]^\top \mathbb{E}_{\boldsymbol{x} \sim P(\boldsymbol{x})} \left[ \boldsymbol{x} \right] \\
&\leq \mathbb{E}_{\boldsymbol{x} \sim P(\boldsymbol{x})} \left[ \boldsymbol{x}^\top \boldsymbol{x} \right] \\
&\leq 1 .
\end{aligned} \tag{37}$$

Given any two different distributions $P(\boldsymbol{x})$ and $Q(\boldsymbol{x})$, define $Z = \int \|P(\boldsymbol{x}) - Q(\boldsymbol{x})\| \, \mathrm{d}\boldsymbol{x}$,[4] we can construct a special probability density function $\widetilde{P}(\boldsymbol{x}) = Z^{-1}\|P(\boldsymbol{x}) - Q(\boldsymbol{x})\|$ to further employee the bounds in (35) and (36). Specifically, for $\gamma \neq 1$, by combining (35) and (37), we can get

$$
\begin{aligned}
(mn)^{-1} \left| \|\boldsymbol{\Theta}_0(P)\|_{\mathrm{tr}} - \|\boldsymbol{\Theta}_0(Q)\|_{\mathrm{tr}} \right| &\overset{(a)}{=} \left| \mathbb{E}_{\boldsymbol{x} \sim P(\boldsymbol{x})} \left[ \boldsymbol{\Theta}_\infty^{(L)}(\boldsymbol{x}, \boldsymbol{x}) \right] - \mathbb{E}_{\boldsymbol{x} \sim Q(\boldsymbol{x})} \left[ \boldsymbol{\Theta}_\infty^{(L)}(\boldsymbol{x}, \boldsymbol{x}) \right] \right| \\
&\overset{(b)}{\leq} \int \|P(\boldsymbol{x}) - Q(\boldsymbol{x})\| \boldsymbol{\Theta}_\infty^{(L)}(\boldsymbol{x}, \boldsymbol{x}) \, d\boldsymbol{x} \\
&\overset{(c)}{\leq} n_0^{-1} Z \|\Sigma_P\|_{\mathrm{tr}} \left( 1 - \gamma^{2L} \right) \left( 1 - \gamma^2 \right)^{-1} \\
&\overset{(d)}{\leq} n_0^{-1} Z \left( 1 - \gamma^{2L} \right) \left( 1 - \gamma^2 \right)^{-1} ,
\end{aligned}
\tag{38}
$$

in which (a) follows from (c) in (34), (b) results from Cauchy Schwarz inequality and (c) is obtained by introducing inequality (35) and distribution $\widetilde{P}(\boldsymbol{x})$. Finally, (d) is based on (37).

Similarly, in the case of $\gamma = 1$, we can get

$$
(mn)^{-1} \left| \|\boldsymbol{\Theta}_0(P)\|_{\mathrm{tr}} - \|\boldsymbol{\Theta}_0(Q)\|_{\mathrm{tr}} \right| \leq n_0^{-1} Z L ,
\tag{39}
$$

which concludes the proof.

**Remark.** Note that ReLU is widely adopted as the nonlinearity in neural networks, which satisfies the Lipschitz continuity with $\gamma = 1$ and the inequality $|\sigma(x)| \leq |x|$. Notably, many other ReLU-type nonlinearities (e.g., Leaky ReLU (Maas et al., 2013) and PReLU (He et al., 2015)) can also satisfy these two conditions.

## APPENDIX B  EXPERIMENTAL SETTINGS

### B.1  DETERMINATION OF CONSTRAINT $\nu$ AND PENALTY COEFFICIENT $\mu$

As demonstrated in Sec. 3.1, constraint $\nu$ (derived from $\|\boldsymbol{\Theta}_0(\mathcal{A})\|_{\mathrm{tr}} < mn\eta^{-1}$ in (8)) introduces a trade-off between the complexity of final selected architectures and the optimization behavior in their model training. This constraint $\nu$ hence is of great importance to our NASI algorithm. Though $\nu$ has already been pre-defined as $\nu \triangleq \gamma n\eta^{-1}$ in Sec. 3.3, we still tend to take it as a hyper-parameter to be determined for the selection of best-performing architectures in practice. Specifically, in this paper, two methods are adopted to determine $\nu$ during the search progress: The **fixed** and **adaptive** method shown as below. Notably, the final architectures selected by NASI via the **fixed** and **adaptive** method are called NASI-FIX and NASI-ADA, respectively. Note that our experiments in the main text suggest that both methods can select architectures with competitive generalization performance over different tasks.

**The fixed determination.**  We initialize and fix $\nu$ with $\nu_0$ during the whole search process. Hence, $\nu_0$ is required to provide a good trade-off between the complexity of architectures and their optimization behavior in the search space. Intuitively, the expectation of architecture complexity in the search space can help to select architectures with medium complexity and hence implicitly achieve a good trade-off between the complexity of architectures and their optimization behavior. Specifically, we randomly sample $N = 50$ architectures in the search space (i.e., $\mathcal{A}_1, \cdots, \mathcal{A}_N$), and then determine $\nu_0$ before the search process by

$$
\nu = \nu_0 = N^{-1} \sum_i^N \|\widetilde{\boldsymbol{\Theta}}_0(\mathcal{A}_i)\|_{\mathrm{tr}} .
\tag{40}
$$

Note that we can further enlarge $\nu_0$ in practice to encourage the selection of architectures with larger complexity.

---

[4] We abuse this integration to ease notations.

**The adaptive determination.** We initialize $\nu$ with a relatively large $\nu_0$ and then adaptively update it with the expected $\|\widetilde{\Theta}_0(\mathcal{A})\|_{\mathrm{tr}}$ of sampled architectures during the search process. Specifically, with sampled architectures $\mathcal{A}_1, \cdots, \mathcal{A}_t$ in the history, $\nu$ at time $t$ (i.e., $\nu_t$) during the search process is given by

$$\nu_t = t^{-1} \left( \nu_0 + \sum_{\tau=1}^{t-1} \|\widetilde{\Theta}_0(\mathcal{A}_\tau)\|_{\mathrm{tr}} \right) . \tag{41}$$

We apply a relatively large $\nu_0$ to ensure a loose constraint on the complexity of architectures in the first several steps of the search process. Note that the adaptive determination provides a more accurate approximation of the expected complexity of architectures in the search space than the fixed determination method if more architectures are sampled to update $\nu_t$, i.e., $t > N$.

Note that (13) in Sec. 3.3 further reveals the limitation on the complexity of final selected architectures by the penalty coefficient $\mu$. Particularly, $\mu=0$ indicates no limitation on the complexity of architectures. Following from the introduced trade-off between the complexity of final selected architectures and the optimization behavior in their model training by the constraint $\nu$, for a search space with relatively larger-complexity architectures, a larger penalty coefficient $\mu$ (i.e., $\mu = 2$) is preferred to search for architectures with relatively smaller complexity to ensure a well-behaved optimization with a larger learning rate $\eta$. On the contrary, for a search space with relatively smaller-complexity architectures, a lower penalty coefficient $\mu$ (i.e., $\mu=1$) is adopted to ensure the complexity and hence the representation power of the final selected architectures. Appendix C.6 provides further ablation study on the constraint $\nu$ and the penalty coefficient $\mu$.

## B.2 THE DARTS SEARCH SPACE

Following from the DARTS (Liu et al., 2019) search space, candidate architecture in our search space comprise a stack of $L$ cells, and each cell can be represented as a directed acyclic graph (DAG) of $N$ nodes denoted by $\{x_0, x_1, \ldots, x_{N-1}\}$. To select the best-performing architectures, we instead need to select their corresponding cells, including the normal and reduction cell. Specifically, $x_0$ and $x_1$ denote the input nodes, which are the output of two preceding cells, and the output $x_N$ of a cell is the concatenation of all intermediate nodes from $x_2$ to $x_{N-1}$. Following that of SNAS (Xie et al., 2019), by introducing the distribution of architecture in the search space (i.e., $p_{\boldsymbol{\alpha}}(\mathcal{A})$ in (14)), each intermediate nodes $x_i$ ($2 \le i \le N - 1$) denotes the output of a single sampled operation $o_i \sim p_i(o)$ with $\sum_{o \in \mathcal{O}} p_i(o) = 1$ given a single sampled input $x_j \sim p_i(x)$ with $j \in 0, \cdots, i-1$ and $\sum_{j=0}^{i-1} p_i(x_j) = 1$, where $\mathcal{O}$ is a predefined operation set. After the search process, only operations achieving the top-2 largest $p_i(o)$ and inputs achieving the top-2 largest $p_i(x)$ for node $x_i$ are retained. Each intermediate node $x_j$ hence connects to two preceding nodes with the corresponding selected operations.

Following from DARTS, the candidate operation set $\mathcal{O}$ includes following operations: $3 \times 3$ max pooling, $3 \times 3$ avg pooling, identity, $3 \times 3$ separable conv, $5 \times 5$ separable conv, $3 \times 3$ dilated separable conv, $5 \times 5$ dilated separable conv. Note that our search space has been modified slightly based on the standard DARTS (Liu et al., 2019) search space: (a) Operation *zero* is removed from the candidate operation set in our search space since it can never been selected in the standard DARTS search space, and (b) the inputs of each intermediate node are selected independently from the selection of operations, while DARTS attempts to select the inputs of intermediate nodes by selecting their coupling operations with the largest weights. Notably, following from DARTS, we need to search for two different cells: A normal cell and a reduction cell. Besides, a max-pooling operation in between normal and reduction cell is applied to down-sampling intermediate features during the search process inspired by NAS-Bench-1Shot1 (Zela et al., 2020b).

## B.3 SAMPLING ARCHITECTURE WITH GUMBEL-SOFTMAX

Notably, aforementioned $p_i(o)$ and $p_i(x)$ for the node $x_i$ follow the categorical distributions. To relax the optimization of (14) with such categorical distributions to be continuous and differentiable, Gumbel-Softmax (Jang et al., 2017; Maddison et al., 2017) is applied. Specifically, supposing $p_i(o)$ and $p_i(x)$ are parameterized by $\alpha_{i,j}^o$ and $\alpha_{i,k}^x$ respectively, with Straight-Through (ST) Gumbel Estimator (Jang et al., 2017), we can sample the single operation and input for the node $x_i$ ($2 \le i \le$

$N - 1$) during the search process by

$$
\begin{aligned}
j^* &= \arg\max_j \frac{\exp((\alpha_{i,j}^o + g_{i,j}^o)/\tau)}{\sum_j \exp((\alpha_{i,j}^o + g_{i,j}^o)/\tau)} \\
k^* &= \arg\max_k \frac{\exp((\alpha_{i,k}^x + g_{i,k}^x)/\tau)}{\sum_k \exp((\alpha_{i,k}^x + g_{i,k}^x)/\tau)} \ ,
\end{aligned}
\tag{42}
$$

where $g_{i,j}^o$ and $g_{i,k}^x$ are sampled from Gumbel(0, 1) and $\tau$ denotes the softmax temperature, which is conventionally set to be 1 in our experiments. Note that in (42), $j \in [0, \cdots, |\mathcal{O}| - 1]$ and $k \in [0, \cdots, i - 1]$, which correspond to the $|\mathcal{O}|$ operations in the operation set $\mathcal{O}$ and the $i$ candidate inputs for the node $x_i$, respectively. Notably, the gradient through ST can be approximated by its continuous counterpart as suggested in (Jang et al., 2017), thus allowing the continuous and differentiable optimization of (14). By sampling the discrete operation and input with (42) for each nodes in the cells, the final sampled architecture can hence be determined.

### B.4 Evaluation on CIFAR-10/100 and ImageNet

**Evaluation on CIFAR-10/100.** Following DARTS (Liu et al., 2019), the final selected architectures consist of 20 searched cells: 18 of them are identical normal cell and 2 of them are identical reduction cell. An auxiliary tower with weight 0.4 is located at 13-th cell of the final selected architectures and the number of initial channels is set to be 36. The final selected architecture are then trained via stochastic gradient descent (SGD) of 600 epochs with a learning rate cosine scheduled from 0.025 to 0, momentum 0.9, weight decay $3 \times 10^{-4}$ and batch size 96 on a single Nvidia 2080Ti GPU. Cutout (Devries & Taylor, 2017), and ScheduledDropPath linearly increased from 0 to 0.2 are also employed for regularization purpose.

**Evaluation on ImageNet.** We evaluate the transferability of the selected architectures from CIFAR-10 to ImageNet. The architecture comprises of 14 cells (12 normal cells and 2 reduction cells). To evaluate in the mobile setting (under 600M multiply-add operations), the number of initial channels is set to 46. We adopt conventional training enhancements (Liu et al., 2019; Chen et al., 2019; Chen & Hsieh, 2020) include an auxiliary tower loss of weight 0.4 and label smoothing. Following P-DARTS (Chen et al., 2019) and SDARTS-ADV (Chen & Hsieh, 2020), we train the model from scratch for 250 epochs with a batch size of 1024 on 8 Nvidia 2080Ti GPUs. The learning rate is warmed up to 0.5 for the first 5 epoch and then decreased to zero linearly. We adopt the SGD optimizer with 0.9 momentum and a weight decay of $3 \times 10^{-5}$ for the model training on ImageNet.

## Appendix C  More Results

### C.1  Trade-off Between Model Complexity and Optimization Behaviour

In this section, we empirically validate the existence of trade-off between model complexity of selected architectures and the optimization behavior in their model training for our reformulated NAS detailed in Sec. 3.1.

**The trade-off in NAS-Bench-1Shot1.** Specifically, Figure 3 illustrates the relation between test error and approximated $\|\mathbf{\Theta}_0(\mathcal{A})\|_{\mathrm{tr}}$ of candidate architectures in the three search spaces of NAS-Bench-1Shot1. Note that with the increasing of the approximated $\|\mathbf{\Theta}_0(\mathcal{A})\|_{\mathrm{tr}}$, the test error decreases rapidly to a minimum and then increase gradually, which implies that architectures only with certain $\|\mathbf{\Theta}_0(\mathcal{A})\|_{\mathrm{tr}}$ (or with desirable complexity instead of the largest complexity) can achieve the best generalization performance. These results hence validate the existence of trade-off between the model complexity and the generalization performance of selected architectures. Note that such trade-off usually results from different optimization behavior in the model training of those selected architectures as demonstrated in the following experiments.

**The trade-off in the DARTS search space.** We then illustrate the optimization behavior of the final selected architecture from the DARTS search space with different constraint and penalty coefficient in Figure 4 to further confirm the existence of such trade-off in our reformulated NAS (8). Specifically,

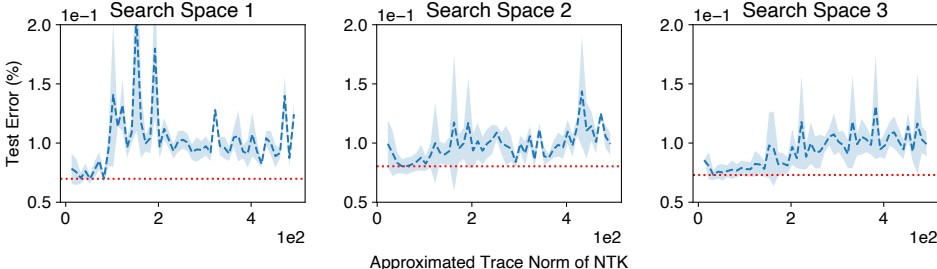

Figure 3: The relation between test error and approximated $\|\Theta_0(\mathcal{A})\|_{\mathrm{tr}}$ of candidate architectures in the three search spaces of NAS-Bench-1Shot1 over CIFAR-10. Note that $x$-axis denotes the approximated $\|\Theta_0(\mathcal{A})\|_{\mathrm{tr}}$, which is averaged over the architectures grouped in the same bin based on their approximated $\|\Theta_0(\mathcal{A})\|_{\mathrm{tr}}$. Correspondingly, $y$-axis denotes the averaged test error with standard deviation (scaled by 0.5) of these grouped architectures. In addition, the red lines demonstrate the smallest test errors achieved by candidate architectures in these three search spaces.

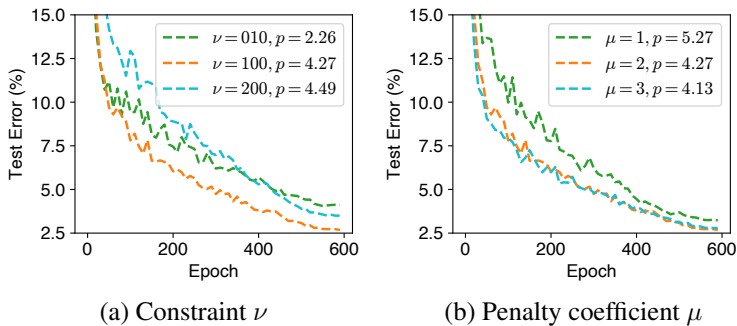

(a) Constraint $\nu$  (b) Penalty coefficient $\mu$

Figure 4: The optimization behavior (test error on CIFAR-10 in model training) of the final selected architectures with different constraint $\nu$ and penalty coefficient $\mu$. The model parameter (MB) denoted by $p$ of each architecture is given in the top-right corner.

we apply NASI with the fixed determination of $\nu$ to select architectures in the the DARTS search space over CIFAR-10, where $\nu_0$ in the fixed determination method introduced in Appendix B.1 is modified manually for the comparison. Besides, a search budget of $T = 100$ and batch size of $b = 64$ are adopted. These final selected architectures are then trained on CIFAR-10 following Appendix 5.2.

Note that, in Figure 4(a), with the increasing of $\nu$, the final selected architectures by NASI contain more parameters and hence achieve larger complexity. Meanwhile, the final selected architecture with $\nu=10$ enjoys a faster convergence rate in the first 50 epochs, but a poorer generalization performance than the one with $\nu= 200$. Interestingly, the final selected architecture with $\nu=100$ realizes the fastest convergence and the best generalization performance by achieving a proper complexity of final selected architectures. These results validate the existence and also the importance of the trade-off between the complexity of final selected architectures and the optimization behavior in their model training. However, the trade-off introduced by the penalty coefficient $\mu$ shown in Figure 4(b) is hard to be observed, which indicates that the constraint $\nu$ is of greater importance than the penalty coefficient $\mu$ in terms of the trade-off between the complexity of architectures and their optimization behavior. Interestingly, Figure 4(b) can still reveal the slower convergence rate and poorer generalization performance caused by the selection of architecture with relatively larger complexity, i.e., $\mu=1$.

### C.2 COMPARISON TO OTHER TRAINING-FREE METRICS

We compare our methods with other training-free NAS methods using both the Spearman correlation and the Kendall's tau between the training-free metrics and the test accuracy on CIFAR-10 in the three search spaces of NAS-Bench-1Shot1. We adopt one uniformly randomly sampled mini-batch data to evaluate these two correlations and apply the same implementations of these training-free

Table 3: Comparison of the Spearman correlation and the Kendall's tau for various training-free metrics in the three search spaces of NAS-Bench-1Shot1 on CIFAR-10.

| Methods | Spearman Correlation | | | Kendall's Tau | | |
|---|---|---|---|---|---|---|
| | S1 | S2 | S3 | S1 | S2 | S3 |
| SNIP (Lee et al., 2019b) | −0.49 | −0.62 | −0.79 | −0.39 | −0.49 | −0.63 |
| GraSP (Wang et al., 2020) | 0.41 | 0.54 | 0.17 | 0.33 | 0.42 | 0.15 |
| SynFlow (Tanaka et al., 2020) | −0.52 | −0.45 | −0.53 | −0.42 | −0.40 | −0.47 |
| NASWOT (Mellor et al., 2021) | 0.21 | 0.32 | 0.54 | 0.16 | 0.24 | 0.44 |
| NASI (conditioned) | **0.62** | **0.74** | **0.76** | **0.44** | **0.53** | **0.53** |

Table 4: The comparison among state-of-the-art (SOTA) NAS algorithms on NAS-Bench-201. The performance of the final architectures selected by NASI is reported with the mean and standard deviation of four independent trials. The search costs are evaluated on a single Nvidia 1080Ti.

| Architecture | Test Accuracy (%) | | | Search Cost (GPU Sec.) | Search Method |
|---|---|---|---|---|---|
| | C10 | C100 | IN-16-120 | | |
| ResNet (He et al., 2016) | 93.97 | 70.86 | 43.63 | - | - |
| ENAS (Pham et al., 2018) | 54.30 | 15.61 | 16.32 | 13315 | RL |
| DARTS (1st) (Liu et al., 2019) | 54.30 | 15.61 | 16.32 | 10890 | gradient |
| DARTS (2nd) (Liu et al., 2019) | 54.30 | 15.61 | 16.32 | 29902 | gradient |
| GDAS (Dong & Yang, 2019) | 93.61±0.09 | 70.70±0.30 | 41.84±0.90 | 28926 | gradient |
| NASWOT (N=10) (Mellor et al., 2021) | 92.44±1.13 | 68.62±2.04 | 41.31±4.11 | 3 | training-free |
| NASWOT (N=100) (Mellor et al., 2021) | 92.81±0.99 | 69.48±1.70 | 43.10±3.16 | 30 | training-free |
| NASWOT (N=1000) (Mellor et al., 2021) | 92.96±0.81 | 69.98±1.22 | 44.44±2.10 | 306 | training-free |
| TE-NAS (Chen et al., 2021) | 93.90±0.47 | 71.24±0.56 | 42.38±0.46 | 1558 | training-free |
| KNAS (Xu et al., 2021) | 93.05 | 68.91 | 34.11 | 4200 | training-free |
| NASI ($T$) | 93.08±0.24 | 69.51±0.59 | 40.87±0.85 | 30 | training-free |
| NASI ($4T$) | 93.55±0.10 | 71.20±0.14 | 44.84±1.41 | 120 | training-free |

NAS methods in (Abdelfattah et al., 2021) for the comparison. Table 3 summarizes the comparison, where the results of our metric are reported under the constraint in (8). Interestingly, our metric generally achieves a higher positive correlation than other training-free metrics, which confirms the reasonableness and also the effectiveness of our training-free metric.

## C.3 SEARCH IN NAS-BENCH-201

To further justify the improved search efficiency and competitive search effectiveness of our NASI algorithm, we also compare it with other state-of-the-art NAS algorithms in NAS-Bench-201 (Dong & Yang, 2020) on CIFAR-10/100 (C10/100) and ImageNet-16-200 (IN-16-200) [5]. Table 4 summarizes the comparison. Note that the baselines in Table 4 are obtained from TE-NAS (Chen et al., 2021) paper. Notably, compared with training-based NAS algorithms, our NASI algorithm can achieve significantly improved search efficiency while maintaining a competitive or even outperforming test performance. Furthermore, our NASI algorithm is shown to be able to enjoy both improved search efficiency and effectiveness when compared with most other training-free baselines. Although TE-NAS, as the best-performing training-free NAS algorithm on both CIFAR-10/100, achieves a relatively improved test accuracy than our NASI ($T$), our NASI with a search budget of $T = 30s$ is $50\times$ more efficient than TE-NAS and is able to achieve compelling test performance on all the three datasets. Moreover, by providing a larger search budget, our NASI algorithm (i.e., NASI ($4T$)) can in fact achieve comparable (on CIFAR-10/100) or even better (on ImageNet-16-120) search results with $12\times$ lower search cost compared with TE-NAS.

## C.4 ARCHITECTURES SELECTED BY NASI

The final selected cells in the DARTS search space (i.e., NASI-FIX and NASI-ADA used in Sec. 5.2) are illustrated in Figure 5, where different operations are denoted with different colors for clarification.

---

[5]ImageNet-16-200 is a down-sampled variant of ImageNet (ImageNet16×16) (Chrabaszcz et al., 2017)

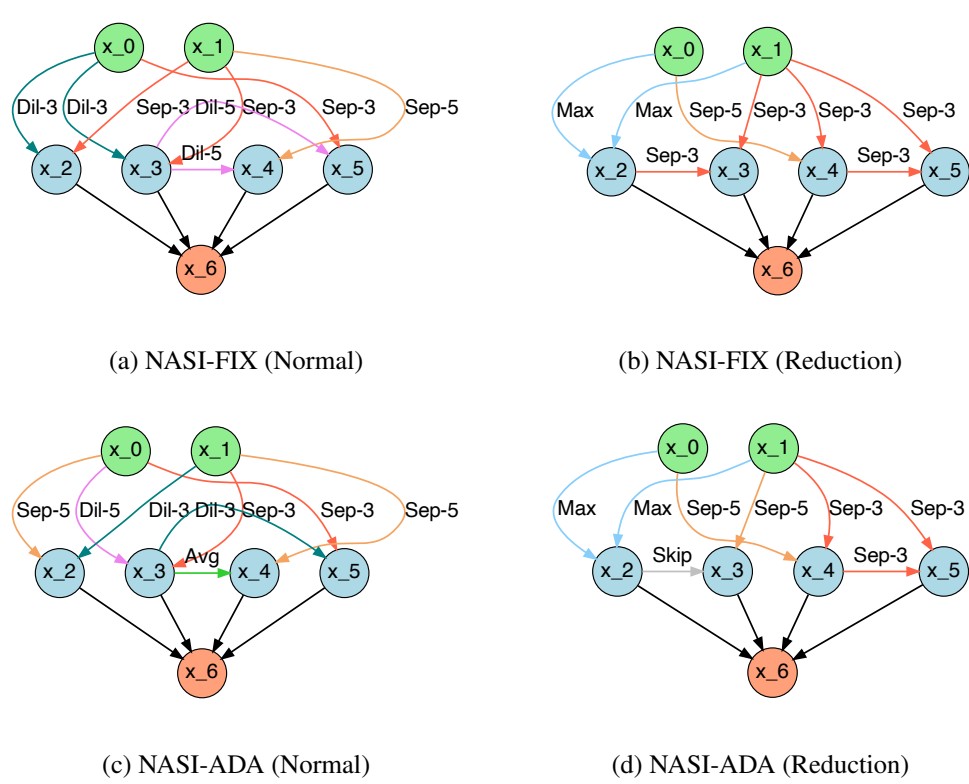

Figure 5: The final selected normal and reduction cells of NASI-FIX and NASI-ADA in the reduced DARTS search space on CIFAR-10. Note that $x_0, x_1$ denote the input nodes, $x_2, x_3, x_4, x_5$ denote intermediate nodes and $x_6$ denotes the output node as introduced in Appendix B.2.

Table 5: Performance comparison among SOTA image classifiers on ImageNet. Note that architectures followed by C10 are transferred from the CIFAR-10 dataset, while architectures followed by ImageNet are directly selected on ImageNet.

| Architecture | Test Error (%) | | Params (M) | $+\times$ (M) | Search Cost (GPU Days) |
|---|---|---|---|---|---|
| | Top-1 | Top-5 | | | |
| Inception-v1 (Szegedy et al., 2015) | 30.1 | 10.1 | 6.6 | 1448 | - |
| MobileNet (Howard et al., 2017) | 29.4 | 10.5 | 4.2 | 569 | - |
| ShuffleNet 2×(v2) (Ma et al., 2018) | 25.1 | 7.6 | 7.4 | 591 | - |
| NASNet-A (Zoph et al., 2018) | 26.0 | 8.4 | 5.3 | 564 | 2000 |
| AmoebaNet-A (Real et al., 2019) | 25.5 | 8.0 | 5.1 | 555 | 3150 |
| PNAS (Liu et al., 2018) | 25.8 | 8.1 | 5.1 | 588 | 225 |
| MnasNet-92 (Tan et al., 2019) | 25.2 | 8.0 | 4.4 | 388 | - |
| DARTS (Liu et al., 2019) | 26.7 | 8.7 | 4.7 | 574 | 4.0 |
| SNAS (mild) (Xie et al., 2019) | 27.3 | 9.2 | 4.3 | 522 | 1.5 |
| GDAS (Dong & Yang, 2019) | 26.0 | 8.5 | 5.3 | 581 | 0.21 |
| ProxylessNAS (Cai et al., 2019) | 24.9 | 7.5 | 7.1 | 465 | 8.3 |
| P-DARTS (Chen et al., 2019) | 24.4 | 7.4 | 4.9 | 557 | 0.3 |
| DARTS- (Chu et al., 2020) | 23.8 | 7.0 | 4.5 | 467 | 4.5 |
| SDARTS-ADV (Chen & Hsieh, 2020) | 25.2 | 7.8 | 5.4 | 594 | 1.3 |
| TE-NAS (C10) (Chen et al., 2021) | 26.2 | 8.3 | 5.0 | - | 0.05 |
| TE-NAS (ImageNet) (Chen et al., 2021) | 24.5 | 7.5 | 5.4 | - | 0.17 |
| NASI-FIX (C10) | 24.3 | 7.3 | 5.2 | 585 | **0.01** |
| NASI-FIX (ImageNet) | 24.4 | 7.4 | 5.5 | 615 | **0.01** |
| NASI-ADA (C10) | 25.0 | 7.8 | 4.9 | 559 | **0.01** |
| NASI-ADA (ImageNet) | 24.8 | 7.5 | 5.2 | 585 | **0.01** |

Interestingly, according to the definitions and findings in (Shu et al., 2020), these final selected cells by NASI are relatively deeper and shallower than the ones selected by other NAS algorithms and hence may achieve worse generalization performance. Nonetheless, in our experiments, the architectures constructed with these cells (i.e., NASI-FIX and NASI-ADA) achieve competitive or even better generalization performance than the ones selected by other NAS algorithms as shown in Table 1 and Table 5. A possible explanation is that our NASI algorithm provides a good trade-off between the complexity of architectures and the optimization in their model training, while other NAS algorithms implicitly prefer architectures with smaller complexity and faster convergence rate as revealed in (Shu et al., 2020). Note that the final selected cells in NASI-FIX and NASI-ADA are of great similarity to each other, which hence implies that the adaptive determination and the fixed determination of constraint $\nu$ share similar effects on the selection of final architectures.

## C.5 EVALUATION ON IMAGENET

We also evaluate the performance of the final architectures selected by NASI on ImageNet and summarize the results in Table 5. Notably, NASI-FIX and NASI-ADA outperform the expert-designed architecture ShuffleNet 2×(v2), NAS-based architecture MnasNet-92 and DARTS by a large margin, and are even competitive with best-performing one-shot NAS algorithm DARTS-. Notably, while achieving better generalization performance than TE-NAS (ImageNet), NASI (C10) is even shown to be more efficient by directly transferring the architectures selected on CIFAR-10 to ImageNet based on its provable transferability in Sec. 4. Meanwhile, by directly searching on ImageNet, NASI-ADA (ImageNet) is able to achieve further improved performance over NASI-ADA (C10) while the performance of NASI-FIX (ImageNet) and NASI-FIX (C10) are quite similar.[6] Above all, these results on ImageNet further confirm the good transferability of the architectures selected by NASI to larger-scale datasets.

---

[6]In order to maintain the same initial channels between NASI-FIX (ImageNet) and NASI-FIX (C10), the multiply-add operations of NASI-FIX (ImageNet) has to be larger than 600M by a small margin.

Table 6: Search with varying architecture widths $N$ in the three search spaces of NAS-Bench-1Shot1.

| Spaces | $N = 2$ | $N = 4$ | $N = 8$ | $N = 16$ | $N = 32$ |
|--------|---------|---------|---------|----------|----------|
| S1 | 7.0±0.6 | 8.3±1.8 | 8.0±2.2 | 7.3±1.5 | 6.5±0.2 |
| S2 | 7.3±0.7 | 8.0±1.5 | 7.3±0.6 | 7.3±0.4 | 7.0±0.2 |
| S3 | 7.6±0.8 | 7.7±1.1 | 6.8±0.1 | 6.8±0.3 | 6.4±0.2 |

Table 7: Pearson correlation between our NTK trace norm approximation and the exact NTK trace norm in the three search spaces of NAS-Bench-1Shot1.

| Spaces | Mini-batch | | | | | | Sum |
|--------|-----------|-----------|------------|------------|------------|-------------|-----|
| | $b = 4$ | $b = 8$ | $b = 16$ | $b = 32$ | $b = 64$ | $b = 128$ | |
| S1 | 0.74 | 0.80 | 0.84 | 0.83 | 0.85 | 0.86 | 0.88 |
| S2 | 0.82 | 0.87 | 0.90 | 0.89 | 0.91 | 0.91 | 0.92 |
| S3 | 0.66 | 0.74 | 0.81 | 0.79 | 0.82 | 0.84 | 0.87 |

## C.6 ABLATION STUDY AND ANALYSIS

**The impacts of architecture width.** As our theoretically grounded performance estimation of neural architectures relies on an infinite width assumption (i.e., $n \to \infty$ in Proposition 1), we investigate the impacts of varying architecture width on the generalization performance of final selected architectures by our NASI. We adopt the same search settings in Sec. 5.1 but with a varying architecture width (i.e., a varying number of initial channels) on CIFAR-10 for the three search spaces of NAS-Bench-1Shot1. Table 6 shows results of the effectiveness of our NASI in NAS-Bench-1Shot1 with varying architecture widths $N$: NASI consistently selects well-performing architectures and a larger width ($N = 32$) enjoys better search results. Hence, the infinite width assumption for NTK does not cause any empirical issues for our NASI.

**The effectiveness of NTK trace norm approximations.** Since the trace norm of NTK is costly to evaluate, we have provided our approximation to it in Sec. 3.2. In this section, we empirically validate the effectiveness of our NTK trace norm approximation. Specifically, we evaluate the Pearson correlation between our approximations (including the approximations using the sum of sample gradient norm in the first inequality of (11) and the approximations using mini-batch gradient norm in (12)) and the exact NTK trace norm under varying batch size in the three search spaces of NAS-Bench-1Shot1. The results are summarized in Table 7. The results confirm that our approximation is reasonably good to estimate the exact NTK trace norm of different architectures by achieving a high Pearson correlation between our approximations and the exact NTK trace norm. Interestingly, a larger batch size of mini-batch gradient norm generally achieves a better approximation, and the sum of sample gradient norm achieves the best approximation, which can be explained by the possible approximation errors we introduced when deriving our (11) and (12).

**The impacts of batch size.** Since we only adopt a mini-batch to approximate the NTK trace norm shown in Sec. 3.2, we further examine the impacts of varying batch size on the search results in the three search spaces of NAS-Bench-1Shot1 in this section. Table 8 summarizes the search results. Interestingly, the results show that our approximation under varying batch sizes achieves comparable search results, further confirming the effectiveness of our approximations in select well-performing architectures.

**The impacts of constraint $\nu$ and penalty coefficient $\mu$.** Based on the analysis in Sec. 3.3 and the results in Appendix C.1, the choice of constraint $\nu$ and penalty coefficient $\mu$ is thus non-trivial to select best-performing architectures since they trade off the complexity of final selected architectures and the optimization in their model training. In this section, we demonstrate their impacts on the generalization performance of the finally selected architectures and also the effectiveness of our fixed determination on the constraint $\nu$ in detail. Notably, we adopt the same settings (including the search and training settings) as those in Appendix C.1 on the DARTS search space, where $\nu_0$ in the fixed determination method introduced in Appendix B.1 is modified manually for the comparison.

Table 8: Search with varying batch sizes $b$ in the three search spaces of NAS-Bench-1Shot1.

| **Spaces** | $b = 8$ | $b = 16$ | $b = 32$ | $b = 64$ | $b = 128$ |
|---|---|---|---|---|---|
| S1 | 6.8±0.3 | 6.7±0.2 | 6.7±0.1 | 6.0±0.3 | 7.0±0.3 |
| S2 | 6.9±0.2 | 7.3±0.4 | 7.1±0.2 | 7.2±0.3 | 6.8±0.2 |
| S3 | 7.0±0.3 | 6.6±0.2 | 6.7±0.2 | 6.6±0.1 | 7.1±0.2 |

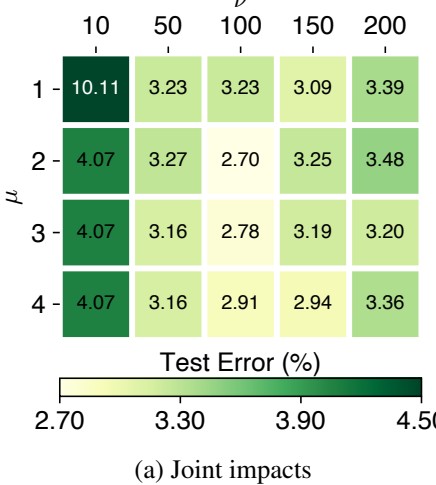

(a) Joint impacts

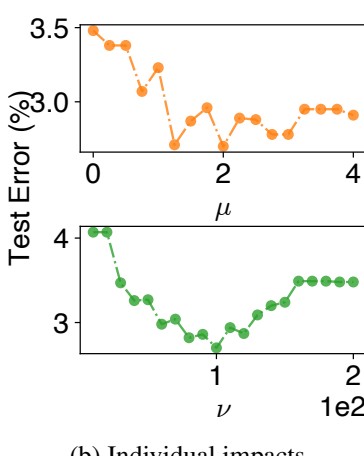

(b) Individual impacts

Figure 6: The impacts of constraint $\nu$ and penalty coefficient $\mu$ on the generalization performance of the final selected architectures by NASI: (a) their joint impacts, and (b) their individual impacts.

Figure 6 summarizes their impacts into two parts: the joint impacts in Figure 6(a) and the individual impacts in Figure 6(b). Notably, in both plots, with the increasing of $\nu$ or $\mu$, the test error of the final selected architecture by NASI gradually decreases to a minimal and then increase steadily, hence further validating the existence of the trade-off introduced by $\nu$ and $\mu$. Moreover, the final selected architectures with $\nu{=}100$ significantly outperform other selected architectures. This result thus supports the effectiveness of our fixed determination method for $\nu$ in NASI. Interestingly, Figure 6(b) reveals a less obvious decreasing trend of test error for $\mu$ compared with $\nu$, which hence further implies the greater importance of constraint $\nu$ than penalty coefficient $\mu$ in terms of the generalization performance of the final selected architectures. Therefore, in our experiments, we conventionally set penalty coefficient $\mu{=}1$ for small-complexity search spaces and $\mu{=}2$ for large-complexity search spaces as introduced in Appendix B.1.

