# OpenReview forum: "NASI: Label- and Data-agnostic Neural Architecture Search at Initialization"
_ICLR.cc/2022/Conference — ICLR 2022 Poster_

### Official Review · Reviewer_PjYZ · 2021-10-21

**Correctness:** 4
**Technical Novelty And Significance:** 3
**Empirical Novelty And Significance:** 3
**Recommendation:** 8
**Confidence:** 4

**Main Review:**

Pros
++ The idea of using NTK as a network performance predictor for NAS is interesting, and the detailed method is non-trivial and plausible (including the relaxation and sampling to solve intractable optimization).
++ The paper is well-written and easy to follow.
++ The paper provides sufficient theoretical proof of its claim, including the assumptions made.
++ The search costs are very appealing and the performance of the searched architectures are good.
++ The proposed method can be data-/label-free, which is a good property that most NAS methods do not have.

Cons
-- The performance of NASI is worse than TE-NAS on NAS-Bench-201 (Table 4).
-- The advantage of NASI on ImageNet is not significant comparing to SOTA (Table 5).
-- How about the results if NASI is directly applied on ImageNet, instead of transferring the best architectures searched on CIFAR-10?

**Summary Of The Paper:**

This paper casts the problem of NAS into a training-free evaluation process by using neural tangent kernel (NTK). Specifically, the paper argues that the training dynamics and the performance of a DNN can be determined by the constant NTK of its linearization. Moreover, to efficiently evaluate the constant NTK of any network architectures, the paper proposes to use  the trac norm of NTK at initialization as an approximation. Using the NAS method proposed in this paper, one can search high-quality architectures with little GPU-hours. Interestigly, the proposed method is robust when applied in a data-/label-free search setting. Extensive experiments show that the searched networks have good performance and can be well-transferred to other datasets.

**Summary Of The Review:**

In general, this paper is of good quality and exceed the bar of being accepted by ICLR. Although I have some minor concerns about the performance, I think the paper has great contributions to the NAS community and should be recommended.

---

> ### Author Response · Authors · 2021-11-20
> **Response to Reviewer PjYZ**
>
> We really appreciate your valuable feedback and constructive comments in improving our paper. We would like to address your concerns below:
>
> ---
>
> > The performance of NASI is worse than TE-NAS on NAS-Bench-201 (Table 4)
>
> We agree with you that NASI is relatively worse than TE-NAS on NAS-Bench-201. However, our search cost is significantly lower than TE-NAS, i.e., 30s vs. 1558s. In light of this, we provide below the search results of NASI with a search cost of 120s to further support the effectiveness of our NASI algorithm. As shown in the table below, our NASI algorithm can in fact achieve comparable (on CIFAR-10/100) or even better (on ImageNet-16-120) search results with $12 \times$ lower search cost compared with TE-NAS. We will add these results to the revised paper.
>
>
> | Method  | C10 Acc (%) | C100 Acc (%) | IN-16-120 Acc (%) | Search Cost (GPU Sec.) |
> | :---        |    :----:   |  :---: | :---: | :---: |
> | TE-NAS      |  93.90 (0.47) | 71.24 (0.56) |  42.38 (0.46) | 1558 |
> | NASI (T) |  93.08 (0.24) | 69.51 (0.59) | 40.87 (0.85) | 30 |
> | NASI (4T) | 93.55 (0.10) | 71.20 (0.14) | 44.84 (1.41) | 120 |
>
> > The advantage of NASI on ImageNet is not significant compared to SOTA (Table 5)
>
> Regarding your concern about Table 5, our results in fact are the results of NASI with a search cost of only **0.24 GPU hours**, which is significantly lower than other baselines, e.g., **4 GPU hours** in TE-NAS (ImageNet) and **200 GPU hours** in ProxylessNAS. Therefore, compared with other NAS algorithms, NASI is superior in achieving **significantly reduced search cost** while maintaining a **competitive test performance**. We thank you for pointing this out, and we will add the information about the search costs of different algorithms in Table 5.
>
>
> > How about the results if NASI is directly applied on ImageNet, instead of transferring the best architectures searched on CIFAR-10?
>
> Regarding your question about applying NASI directly on ImageNet, we are in the process of getting the results. Unfortunately, it will take us a few weeks to get them, which may not be ready before the deadline of this discussion. Therefore, we will add these results in our revised paper. As our transferred architectures can already achieve competitive performance on ImageNet (Table 5), we expect that the final selected architectures on ImageNet will also achieve competitive (or even improved) performance.
>
> ---
>
> We thank you again for appreciating our contributions. We sincerely hope our clarifications above have addressed your concerns.

---

> > ### Comment · Reviewer_PjYZ · 2021-11-26
> > **Post-rebuttal**
> >
> > Thank you for the explanation and supportive results on my concerns. I am satisfied with the first two responses. However, it would be better if there are some results on the direct-search-on-ImageNet experiment.
> > Overall, I will keep my initial rating.

---

> > > ### Author Response · Authors · 2021-11-27
> > > **Thank you for appreciating our response!**
> > >
> > > We like to thank you for appreciating our explanation and supportive results on your concerns. We will post the results on the direct-search-on-ImageNet experiment as soon as when they are ready.

---

### Official Review · Reviewer_DrxZ · 2021-11-01

**Correctness:** 3
**Technical Novelty And Significance:** 3
**Empirical Novelty And Significance:** 3
**Recommendation:** 6
**Confidence:** 5

**Main Review:**

Strengths:
The targeting problem of searching neural architectures at initialization is important and promising.
The paper finds that such NASI is guaranteed to be label- and data-agnostic under mild conditions, which demonstrates that the searched architecture can be transferred to different datasets or tasks.
The retrained accuracy is promising and comparable or even better than other gradient-based searching methods.


Weakness:
The authors claim competitive effectiveness on both CIFAR and ImageNet, while I only find CIFAR results in Table 1 and ImageNet-16-200 in Table 4 (also 16 and 200 are not defined).
How comparable are the achieved accuracy to other NAS methods, e.g., FBNetv3, EfficientNet?


**Summary Of The Paper:**

This work proposes to search for good candidate neural architectures at initialization (NASI) so that we can completely avoid model training during the search.

**Summary Of The Review:**

This work proposes to search for good candidate neural architectures at initialization (NASI) so that we can completely avoid model training during the search. The theoretical analysis is also provided to analyze the optimization via NTK tools.

The authors replied my previous major concern with clarity, so I raised my score to 6.

---

> ### Author Response · Authors · 2021-11-20
> **Response to Reviewer DrxZ**
>
> We thank you for your valuable feedback and like to address your concerns below:
>
> ---
>
> ImageNet-16-200, adopted from the NAS-Bench-201 paper, is a down-sampled variant of ImageNet (ImageNet16×16). We thank you for pointing out the missing description of this dataset, and we will provide the description and citation for this dataset in our revised paper. In fact, the results of our NASI on ImageNet are already reported in Appendix C.5 (Table 5) where our algorithm can indeed achieve competitive search results compared with other NAS baselines. The table below further summarizes the comparison with the other baselines (e.g., FBNetv3, EfficientNet) that you have mentioned on ImageNet. Notably, compared with FBNetv3 and EfficientNet, our method can significantly reduce the search cost (by at least $5 \times 10^4$ times) while achieving compelling test accuracies. We think that these results are already very appealing for **training-free NAS**. As FBNetv3 searches for both architecture and training hyper-parameters jointly, it achieves the highest test accuracies among these three methods. Instead, our NASI only searches for architecture and adopt the same training settings as other NAS baselines in order to provide a fair comparison. Searching for both architecture and hyper-parameters jointly in a semi-training-free manner will be an interesting direction for our future work. We will add the above discussion in our revised paper.
>
>
> | Method  | Top-1 Acc (%) | Top-5 Acc (%) | Search Cost (GPU Hours) |
> | :---        |    :----:   |  :---: | :---: |
> | EfficientNet-B0 | 77.1 | 93.3 | >91000 |
> | FBNetV3      | 79.1 | 94.5 | 10700 |
> | NASI-FIX | 75.7 | 92.7 | **0.24** |
>
> ---
>
> We sincerely hope our clarifications above have addressed your concerns and can further improve your opinion of our work.

---

> ### Author Response · Authors · 2021-11-23
> **Thank you for raising your score!**
>
> We are glad to know that our response has addressed your major concern with clarity, as mentioned in your updated summary of review. We also like to thank you for raising your score for our paper.

---

### Official Review · Reviewer_2dLC · 2021-11-02

**Correctness:** 3
**Technical Novelty And Significance:** 3
**Empirical Novelty And Significance:** 3
**Recommendation:** 6
**Confidence:** 4

**Main Review:**

Strengths:
Training-free NAS method is interesting and arouses a lot of interests in NAS field. Based on the unchanged characteristic of NTK, this paper utilizes NTK to evaluate the candidate architectures at initialization. NASI is well-supported by the theorem of NTK and achieves competitive results empirically.

Weakness:
The main concern is the approximations in the paper. Since NTK has a certain assumption about the neural network, the unchanged characteristic may not be satisfied for some neural architectures. To calculate the trace norm of NTK and to solve the NAS problem efficiently, the authors both apply approximations here. Since three approximations are applied, the final results may have a large deviation.

Besides, I also have the following questions about the paper
1. NASI is based on the theory of NTK, which has the infinity-width assumption. Also, there are approximations in Section 3.2 and Section 3.3. Can you discuss the bound of the approximation? Or just verify it empirically?
2. In Proposition 1, you discuss the upper bound of the training loss. However, when the architecture convergences, we focus on the minimum value of the training loss. I am confused about the claim “we can simply minimize the upper bound of $\mathcal{L}_t$” below the Proposition 1.
3. There are some other training-free NAS methods (e.g., [1]). Can you compare NASI with them in a fair manner?

[1] Mellor, Joe, et al. "Neural architecture search without training." International Conference on Machine Learning, 2021.


**Summary Of The Paper:**

This paper proposes a training-free NAS method called NASI, which exploits the Neural Tangent Kernel (NTK) to characterize the performance of the candidate architectures at initialization. To alleviate the costly evaluation for NTK, the authors apply a similar form to gradient flow to approximate NKT. Moreover, they combined their NTK trick with gradient-based NAS algorithm via Gumbel-Softmax to solve NAS problem efficiently. The experiment results on various benchmarks illustrate the effect of NASI.

**Summary Of The Review:**

NASI applies approximations for neural architecture search at initialization based on NTK. However, some of them are not convincing to me (see the main review above). If the authors can provide detailed explanations about that, I would consider raising my score.

---

> ### Author Response · Authors · 2021-11-20
> **Response to Reviewer 2dLC**
>
> We thank you for your valuable feedback in improving our paper. We would like to address your concerns below:
>
> > The main concern is the approximations in the paper. Since NTK has a certain assumption about the neural network, the unchanged characteristic may not be satisfied for some neural architectures. To calculate the trace norm of NTK and to solve the NAS problem efficiently, the authors both apply approximations here. Since three approximations are applied, the final results may have a large deviation...Can you discuss the bound of the approximation? Or just verify it empirically?
>
> Thank you for pointing out such a concern. To simplify the analysis in our paper, we borrow the infinite-width assumption in [R1] to get an unchanged NTK during training. In fact, recent works have shown that this unchanged NTK only requires large finite-width DNNs. Such a condition usually can be well satisfied by over-parameterized DNNs in practice and therefore real-world DNNs of finite width can also approximately enjoy this unchanged NTK, as supported both empirically [R2] and theoretically [R3]. Moreover, we have provided an ablation study in Table 6 to show that our NASI algorithm will select well-performing architectures consistently under varying finite architecture widths, and a larger width generally enjoys a better search result, which aligns with the findings in [R2, R3] where a smaller width will cause a larger approximation error. Above all, the infinite-width assumption for NTK would not be a concern in our NASI algorithm.
>
> To the best of our knowledge, it is non-trivial to give a bound of our approximations to the NTK trace norm. Instead, we have provided empirical studies in Tables 7 and 8 to show the effectiveness of our approximations. Remarkably, Table 7 shows that "our approximation is reasonably accurate in estimating the exact NTK trace norm of different architectures by achieving a high Pearson correlation between our approximations and the exact NTK trace norm" under varying batch sizes. Moreover, Table 8 suggests that "our approximation under varying batch sizes achieves comparable search results," which further confirms "the effectiveness of our approximations in selecting well-performing architectures." So, the approximations to the NTK trace norm would also not be a concern in our NASI algorithm.
>
> [R1] Ziwei Ji, Matus Telgarsky, and Ruicheng Xian. Neural Tangent Kernels, Transportation Mappings, and Universal Approximation. arXiv:1910.06956, 2019.
>
> [R2] Jaehoon Lee, Lechao Xiao, Samuel S. Schoenholz, Yasaman Bahri, Roman Novak, Jascha Sohl-Dickstein, and Jeffrey Pennington. Wide neural networks of any depth evolve as linear models under gradient descent. In *Proc. NeurIPS*, 2019.
>
> [R3] Sanjeev Arora, Simon S. Du, Wei Hu, Zhiyuan Li, Ruslan Salakhutdinov, and Ruosong Wang. On Exact Computation with an Infinitely Wide Neural Net. arXiv:1904.11955, 2019.
>
> > I am confused about the claim “we can simply minimize the upper bound of” below the Proposition 1.
>
> If we understand your question correctly, you are asking why minimizing the upper bound of $\mathcal{L}_t$ can also minimize $\mathcal{L}_t$ itself. Since our upper bound of $\mathcal{L}_t$ is an approximation of $\mathcal{L}_t$, minimizing this upper bound can also approximately minimize $\mathcal{L}_t$. In fact, such an optimization trick of minimizing an upper bound (or maximizing a lower bound) has been widely adopted in optimization. For example, maximizing the model evidence in variational inference can be converted into maximizing an evidence lower bound (ELBO), while minimizing a convex function with constraints can be converted into maximizing its lower bound (i.e., the Lagrangian function). Moreover, our NASI using this trick has indeed found architectures with compelling performance, which therefore verifies the effectiveness of this optimization trick.
>
>
> > There are some other training-free NAS methods (e.g., [1]). Can you compare NASI with them in a fair manner?
>
> We have provided a fair comparison to [1] (i.e., NASWOT) and also other training-free NAS baselines in Table 4. The results show that our NASI can consistently enjoy both improved search efficiency (search cost) and effectiveness (test performance) on various benchmark datasets.
>
> ---
>
> We sincerely hope our clarifications above have addressed your concerns and can further improve your opinion of our work.

---

> > ### Comment · Reviewer_2dLC · 2021-11-25
> > **Post-rebuttal**
> >
> > I think the comparison in Table 4 is not fair since NASWOT costs 4.8 GPU Sec. while NASI costs 30 GPU Sec. However, my main concerns of approximations have been addressed and I decide to lift the score to 6.

---

> > > ### Author Response · Authors · 2021-11-26
> > > **Thank you for increasing your score!**
> > >
> > > We are glad to know that our response has addressed your main concerns of approximations. Regarding a fair comparison with NASWOT, we further provide the search results of NASWOT based on a search budget of 30 and 300 GPU Sec. in the table below. The results show that even though our NASI (T) can only achieve competitive (or slightly worse) performance compared with NASWOT (N=100) given a search budget of 30 GPU Sec., our NASI (4T) can consistently achieve an improved performance that is better than that of NASWOT (N=1000) at a much lower search cost. We will add these results and the above discussion to our revised paper.
> > >
> > > | Method  | C10 Acc (%) | C100 Acc (%) | IN-16-120 Acc (%) | Search Cost (GPU Sec.) |
> > > | :---        |    :----:   |  :---: | :---: | :---: |
> > > | NASWOT (N=100)      |  92.81 (0.99) | 69.48 (1.70) |  43.10 (3.16) | 30 |
> > > | NASWOT (N=1000)      |  92.96 (0.81) | 69.98 (1.22) |  44.44 (2.10) | 306 |
> > > | NASI (T) |  93.08 (0.24) | 69.51 (0.59) | 40.87 (0.85) | 30 |
> > > | NASI (4T) | 93.55 (0.10) | 71.20 (0.14) | 44.84 (1.41) | 120 |
> > >
> > > We sincerely hope our additional clarification has addressed your remaining concern.

---

> ### Comment · Area_Chair_eF48 · 2021-11-25
> **Please provide feedback**
>
> Dear Reviewer 2dLC,
>
> Authors have provided detailed responses to your comments. Please go over them and provide feedback.
>
> Thanks,
> Area Chair

---

### Official Review · Reviewer_iQzj · 2021-11-02

**Correctness:** 3
**Technical Novelty And Significance:** 4
**Empirical Novelty And Significance:** 3
**Recommendation:** 6
**Confidence:** 3

**Main Review:**

+ The proposed method does not require the model training for architecture search, thus it is much more efficient than the previous NAS method.
+ In addition to the efficiency, the proposed method can adapt to the label- and data-agnostic situations.

Concerns:
- Although NTK assumes infinite-depth DNNs, the proposed method does not seem to meet the condition. It would be nice to provide the reason why the proposed method can work. Also, I am interested in the lower bound of the network depth where the proposed method can work well.

- The main concern about this method is whether the proposed method finds good architectures. Firstly, recent studies have claimed that NAS methods find architectures showing good performance but the rank of the found architecture is far from the best [Yu+, ICLR'20]. This comes from the fact that many candidates of CNN architectures in the search space can achieve good performance on the benchmark dataset such as CIFAR-10. In that sense, even though the proposed method achieves good performance on the benchmark datasets, I am wondering if the found architectures are really good or not. To check the rank of the architectures, it would be nice to use NAS-Bench-101 [Ying+, ICML'19]. Secondly, a recent study has argued that training protocols (e.g. data augmentation, training epochs) affect more on performance rather than search algorithms [Yang+, ICLR'20]. In fact, the training protocol affects a lot on the performance of DNNs, so I am wondering if the training protocol affects the estimation of the performance of candidate architectures by the proposed method or not.

- The proposed method is inferior to another training-free method according to Table 4 in the appendix but shows better in the main paper. It would be nice to provide its cause and deeper analysis to better understand the proposed method.

- Is it possible to directly apply the proposed method to a large dataset such as ImageNet?

[Yu+, ICLR'20] Evaluating the Search Phase of Neural Architecture Search, ICLR'20.

[Ying+, ICML'19] NAS-Bench-101: Towards Reproducible Neural Architecture Search, ICML'19.

[Yang+, ICLR'20] NAS evaluation is frustratingly hard

**Summary Of The Paper:**

This paper proposes a new training-free NAS method, where it is not necessary to optimize the weight parameters of target networks for architecture search. To achieve this, the paper exploits the capability of NTK for estimating the performance of candidate architectures at weight initialization. Thus, the proposed method can avoid network training during the search and achieve a much efficient architecture search. The experimental results show that the proposed method achieves competitive performance with existing methods and also can adapt to the label- and data-agnostic scenarios.

**Summary Of The Review:**

The proposed method shows promising results. My major concern is about the empirical and theoretical analysis (see concerns above). Hopefully, the authors can address my concern in the rebuttal period.

---

> ### Author Response · Authors · 2021-11-20
> **Response #2 to Reviewer iQzj**
>
> > The proposed method is inferior to another training-free method according to Table 4 in the appendix but shows better in the main paper. It would be nice to provide its cause and deeper analysis to better understand the proposed method.
>
> Different from pruning-based TE-NAS, our sampling-based NASI needs to traverse as many architectures as possible (i.e., a large $T$) in order to characterize a good distribution of candidate architectures, where the best-performing architecture can be obtained by maximizing over this distribution. Therefore, the performance of our NASI will be improved when a larger $T$ is adopted. In Table 1 of our main paper, we have applied our NASI algorithm with a budget of 0.24 GPU hours (vs. 0.5 GPU hours in TE-NAS), which is sufficient for NASI to achieve compelling performance. However, in Table 4, we only provide $\sim 50 \times$ smaller budget compared with TE-NAS (i.e., 30s vs. 1558s) in order to justify the improved efficiency of our NASI algorithm, which may be too small for NASI to achieve more superior performance. In light of this, we provide below the search results of NASI with a search cost of 120s to further support the effectiveness of our NASI algorithm. As shown in the table below, our NASI algorithm can in fact achieve comparable (on CIFAR-10/100) or even better (on ImageNet-16-120) search results with $12 \times$ lower search cost compared with TE-NAS. We will add these results to the revised paper.
>
> | Method  | C10 Acc (%) | C100 Acc (%) | IN-16-120 Acc (%) | Search Cost (GPU Sec.) |
> | :---        |    :----:   |  :---: | :---: | :---: |
> | TE-NAS      |  93.90 (0.47) | 71.24 (0.56) |  42.38 (0.46) | 1558 |
> | NASI (T) |  93.08 (0.24) | 69.51 (0.59) | 40.87 (0.85) | 30 |
> | NASI (4T) | 93.55 (0.10) | 71.20 (0.14) | 44.84 (1.41) | 120 |
>
>
> > Is it possible to directly apply the proposed method to a large dataset such as ImageNet?
>
> Regarding your question about applying NASI directly on ImageNet, we are in the process of getting the results. Due to the high computational cost of training DNNs on ImageNet and also our limited GPU resources, it will take us a few weeks before we can reach a final conclusion. Therefore, we will add these results in our revised paper. As our transferred architectures can already achieve competitive performance on ImageNet (Table 5), we expect that the final selected architectures on ImageNet will also achieve competitive (if not, better) performance.
>
> ---
> We sincerely hope our clarifications above have addressed your concerns and can further improve your opinion of our work.

---

> ### Author Response · Authors · 2021-11-20
> **Response #1 to Reviewer iQzj**
>
> We really appreciate your valuable and constructive feedback in improving our paper. We would like to address your concerns below:
>
> ---
>
> > Although NTK assumes infinite-depth DNNs, the proposed method does not seem to meet the condition. It would be nice to provide the reason why the proposed method can work. Also, I am interested in the lower bound of the network depth where the proposed method can work well.
>
> If we understand correctly, you are referring to the assumption of infinite-**width** DNNs and the lower bound of the network **width**. To simplify the analysis in our paper, we borrow the infinite-width assumption in [R1] for NTK. In fact, recent works have shown both empirically  [R2] and theoretically [R3] that the conclusions for NTK still hold for large finite-width DNNs. Therefore, our results can also be applied to real-world DNNs of finite width. To the best of our knowledge, it is non-trivial to provide a theoretical lower bound of the network width where our method can work well. Instead, we have provided an ablation study in Table 6 to show that our NASI algorithm can consistently select well-performing architectures under varying finite architecture widths (even for $N=2$), and a larger width generally enjoys a better search result, which aligns with the findings in [R2, R3] where a smaller width will cause a larger approximation error when using NTK at initialization. Given these theoretical and empirical evidence, the infinite-width assumption for NTK would not be a concern in our NASI algorithm.
>
> [R1] Ziwei Ji, Matus Telgarsky, and Ruicheng Xian. Neural Tangent Kernels, Transportation Mappings, and Universal Approximation. arXiv:1910.06956, 2019.
>
> [R2] Jaehoon Lee, Lechao Xiao, Samuel S. Schoenholz, Yasaman Bahri, Roman Novak, Jascha Sohl-Dickstein, and Jeffrey Pennington. Wide neural networks of any depth evolve as linear models under gradient descent. In *Proc. NeurIPS*, 2019.
>
> [R3] Sanjeev Arora, Simon S. Du, Wei Hu, Zhiyuan Li, Ruslan Salakhutdinov, and Ruosong Wang. On Exact Computation with an Infinitely Wide Neural Net. arXiv:1904.11955, 2019.
>
>
> > The main concern about this method is whether the proposed method finds good architectures... so I am wondering if the training protocol affects the estimation of the performance of candidate architectures by the proposed method or not.
>
> Thank you for pointing out these concerns. We'd like to answer your concerns below:
>
> 1. As suggested, we provide below the ranks (i.e., the top percentage) of the architectures found by our NASI in NAS-Bench-1Shot1 and NAS-Bench-201 (to be consistent with the experiments in our main paper). The results show that our method can indeed find good architectures that are ranking high in different search spaces and datasets.
>
> | NAS-Bench-1Shot1  | S1 | S2 | S3 |
> | :---        | :----: |  :---: | :---: |
> | Random      | 32.7% | 28.3% |  13.4% |
> | ENAS      | 11.4% | 25.7% |  30.5% |
> | DARTS      | 11.4% | 4.8% |  3.7% |
> | NASI      | **5.6%** | **4.8%** |  **1.4%** |
>
> | NAS-Bench-201  | C10 | C100| IN-16-120 |
> | :---        | :----: |  :---: | :---: |
> | NASWOT | 36.1% | 36.5% | 46.4% |
> | TE-NAS | 0.5% | 0.8% | 12.6% |
> | KNAS | 8.5% | 15.2% | 58.0% |
> | NASI     | 2.5% | 2.3% |  **1.7%** |
>
>
> 2. Our results on NAS-Bench-1Shot1, NAS-Bench-201, and DARTS search space (evaluated under different settings) have indicated that NASI algorithm can also find well-performing architectures with significantly improved search efficiency, even when DNNs are trained with **different training protocols** used by different search spaces and datasets. These results therefore imply that training protocols likely have **very limited** impact on our NASI algorithm.

---

> > ### Comment · Reviewer_iQzj · 2021-11-22
> > **Responses to the authors**
> >
> > Thank you so much for the detailed responses. First of all, I was mentioning the infinite-depth in the review, but it was my mistake and it should be infinite-width as you pointed out.
> >
> > Although my concerns have mostly been addressed, one remaining concern is how the proposed method considers different training protocols. The authors tested their method on several benchmarks and showed that the proposed method achieved good performance, which indicates that the proposed method is robust to different search spaces and training protocols.
> >
> > However, my question is whether the proposed method (or the infinite-width assumption) can ignore the influence of different training protocols. For instance, I'd say that we would get different performances and the ranking would change when using different training protocols (e.g. different learning rates, scheduling of them, data augmentation) on the same search space. In other words, the ranking of architectures depends on training protocols as well. However, it seems impossible to take these aspects into account at the initialization phase. I am wondering how the proposed method deals with the influence of different training protocols. Also, I am wondering if the infinite-width assumption can ignore the influence of the training protocols.
> >
> > Hopefully, the authors can address my concern.

---

> > > ### Author Response · Authors · 2021-11-22
> > > **Regarding your remaining concern**
> > >
> > > Thank you for appreciating our detailed responses. We would like to address your remaining concern below:
> > >
> > > We agree with you that it is impossible to take most training protocols into account at the initialization phase of DNNs. Therefore, our proposed method based on initialization and infinite-width assumption would not be able to reflect the **subtle change** of architectures' ranking when different training protocols are applied, which is also a **common problem** in training-free NAS. However, in the literature, it has been widely shown that a high-quality architecture can consistently achieve better performance than a low-quality one, even when trained with different training protocols (e.g., ResNet vs. VGG). In light of this, the high-quality architectures selected by our training-free method should be able to **consistently achieve compelling performance**, even when different training protocols are applied (i.e., the **robustness** of our method), which indeed has been well supported by our empirical results on the NAS-Bench-1Shot1, NAS-Bench-201, and DARTS search space. As for NAS that is aware of different training protocols, a joint search of architecture and hyper-parameters should provide a good solution, which is outside the scope of our paper but would constitute an interesting and promising future research direction for training-free NAS.
> > >
> > > We sincerely hope our additional clarifications have addressed your remaining concern and can improve your evaluation of our work.

---

> ### Author Response · Authors · 2021-11-27
> **Regarding our clarifications**
>
> Dear Reviewer iQzj,
>
> We sincerely hope that our clarifications below have well addressed all your concerns and can improve your opinion of our paper. If you have any other remaining concern, we'd like to address it before the deadline of this discussion period (29 Nov). Thanks!

---

### Comment · Area_Chair_eF48 · 2021-11-20
**Rebuttal deadline approaching soon**

Dear authors,

The end of the author response period is quickly approaching (It is November 22nd). Please provide responses to the reviewer's comments and upload the revised version of the paper, since you can no longer edit your paper after the deadline.

Thanks,
Area Chair

---

### Decision · Program_Chairs · 2022-01-20

**Decision:**

Accept (Poster)

**Comment:**

This paper proposes an efficient training-free NAS method, NASI, which exploits Neural Tangent Kernels (NTK)’s ability to estimate the performance of candidate architectures. Specifically, the authors provide a theoretical analysis showing that NAS can be realizable at initialization, and propose an efficient approximation of the trace norm of NTK that has a similar form to gradient flow, to alleviate the prohibitive cost of computing NTK. Since the method is training-free, NASI is also label- and data-agnostic. The experimental validation shows that NASI either outperforms or performs comparably to existing training-based and training-free NAS methods, while being significantly more efficient.

The below is the summary of pros and cons of the paper, after :

Pros
- The idea of using NTK to predict the performance of candidate neural architectures is both novel and promising, and the proposed analysis and efficient approximation are non-trivial.
- The paper provides sufficient theoretical proof of its claims, including the assumptions made.
- The method is highly efficient in terms of search cost, and the searched architectures obtain good performance on benchmark datasets.
- The method is data/label free and thus allows transfer architectures across tasks.
- The paper is well-written.

Cons
- There is no result on ImageNet obtained by directly applying NASI on it.

The initial reviews were split, due to other concerns regarding whether the proposed method finds good architectures, missing comparison against certain training-free baselines, and some unclear descriptions. However, they were addressed away by the authors during the rebuttal period which led to a consensus to accept the paper.

In sum, this is a strong paper that proposes a novel idea for training-free NAS, and the proposed method seems to be both effective, efficient, and generalizes well across tasks. One remaining concern is the computational cost of running the method on larger datasets, such as ImageNet, and I suggest the authors report the results and the running time in the final paper.

Another suggestion is to include discussion of, or comparison to other efficient NAS methods based on meta-learning, such as MetaD2A [Lee et al. 21], which is not training-free but is more efficient than the proposed NASI.

[Lee et al. 21] Rapid Neural Architecture Search by Learning to Generate Graphs from Datasets, ICLR 2021